

# Impact of metabolic pathways and salinity on the hydrogen isotope ratios of haptophyte lipids

Gabriella M. Weiss[1], David Chivall[1,a], Sebastian Kasper[1], Hideto Nakamura[2,3,b], Fiz da Costa[4,c], Philippe Soudant[4], Jaap S. Sinninghe Damsté[1,5], Stefan Schouten[1,5], Marcel T. J. van der Meer[1]

[1]Department of Marine Microbiology and Biogeochemistry, NIOZ Royal Netherlands Institute for Sea Research and Utrecht University, Den Burg, The Netherlands
[2]Department of Earth and Planetary Sciences, Faculty of Science, Hokkaido University, N10W8, Kita-ku, Sapporo 060-0810, Japan
[3]CREST, Japan Science and Technology Agency (JST), N10W8, Kita-ku, Sapporo 060-0810, Japan
[4]Laboratoire des Sciences de l'Environnement Marin (LEMAR UMR 6539CNRS/UBO/IRD/Ifremer), IUEM, rue Dumont d'Urville, 29280 Plouzané, France
[5]Department of Earth Sciences, Faculty of Geosciences, Utrecht University, Utrecht, The Netherlands

Present Addresses:

[a]Oxford Radiocarbon Accelerator Unit, University of Oxford, Oxford OX1 3TG, United Kingdom
[b]Department of Biology and Geosciences, Osaka City University, Sugimoto 3-3-138, Sumiyoshi-ku, Osaka, 558-8585, Japan
[c]Centro de Investigación Mariña, Universidade de Vigo, Illa de Toralla s/n, 36331, Vigo, Spain

*Correspondence to*: Gabriella M. Weiss (gabriella.weiss@nioz.nl)

**Abstract.** Hydrogen isotope ratios of biomarkers have been shown to reflect water isotope ratios, and in some cases correlate significantly with salinity. The $\delta^2$H-salinity relationship is best studied for long-chain alkenones, biomarkers for haptophyte algae, and is known to be influenced by a number of different environmental parameters. It is not fully known why $\delta^2$H ratios of lipids retain a correlation to salinity, and whether this is a general feature for other lipids produced by haptophyte algae. Here, we analyzed $\delta^2$H ratios of three fatty acids, brassicasterol, long-chain $C_{37}$ alkenones and phytol from three different haptophyte species grown over a range of salinities. Lipids synthesized in the cytosol, or relying on precursors of cytosolic origin, show a correlation between their $\delta^2$H ratios and salinity. In contrast, biosynthesis in the chloroplast, or utilizing precursors created in the chloroplast, yields lipids that do not show a correlation between $\delta^2$H ratios and salinity. This leads to the conclusion that location of metabolism is the first-order control on the salinity signal retained in $\delta^2$H ratios of certain lipids. Additionally, we found that $\delta^2$H ratios of alkenones from batch cultures of the Group II haptophyte species *Tisochrysis lutea* correlate positively with temperature, contrary to findings from cultures of Group III haptophytes, but retain a similar response to nutrient availability in line with other Group III haptophytes.

## 1 Introduction

One of the best studied lipids for hydrogen isotope fractionation are long-chain alkenones, which are ethyl and methyl ketones with chain lengths of 35 - 42 carbon atoms and two to four degrees of unsaturation, found globally in lakes and oceans (De Leeuw et al., 1980; Volkman et al., 1980; Cranwell, 1985). Long-chain alkenones are synthesized by specific haptophyte algae

from the order Isochrysidales, and are believed to be created from fatty acids via chain elongation and subsequent decarboxylation (Rontani et al., 2006). Haptophytes are genetically divided into three distinct groups: Groups I and II encompass haptophytes preferring lacustrine and coastal environments (although some Group II haptophytes can also be found in hypersaline lakes), and Group III comprises marine alkenone producers (Theroux et al., 2010). Hydrogen isotope ratios of $C_{37}$ alkenones ($\delta^2 H_{C37}$) and fractionation factor ($\alpha_{C37}$) values both have a significant positive correlation to salinity in cultures

of Group II and Group III haptophytes (Schouten et al., 2006; M'Boule et al., 2014; Chivall et al., 2014; Sachs et al., 2016; Weiss et al., 2017); Group I haptophytes are presently uncultured and thus nothing is known about how they fractionate hydrogen. $\delta^2 H_{C37}$ ratios have been applied to reconstruct salinity across different timescales and marine environments (Pahnke et al., 2007; Leduc et al., 2013; Kasper et al., 2014, 2015; Petrick et al., 2015; Simon et al., 2015; Weiss et al., *in review*). In addition to alkenones, a strong correlation between $\delta^2 H$ ratios and salinity has also been shown for fatty acids and sterols from

haptophytes and other phytoplankton in laboratory cultures (e.g. Sachs and Kawka, 2015; Sachs et al., 2016; Maloney et al., 2016), and is also found for other biomarkers such as $C_{17}$ n-alkane, diploptene, and phytene (see compilation in Sachse et al., 2012).

Despite the strong relationship noted between $\delta^2 H$ ratios and salinity, the precise intracellular mechanisms controlling the

salinity relationship are not well constrained. Furthermore, a number of extracellular factors have been shown to affect $\alpha_{biomarker}$ and $\delta^2 H_{biomarker}$ ratios in culture (Schouten et al., 2006; Wolhowe et al., 2009; Chivall et al., 2014; M'Boule et al., 2014; Sachs and Kawka, 2015; van der Meer et al., 2015; Sachs et al., 2016; Maloney et al., 2016; Weiss et al., 2017). For example, growth rate is shown to negatively correlate with $\delta^2 H_{C37}$ ratios and $\alpha_{C37}$ values in Group III haptophyte species and is thought to be a limiting factor for reconstructions of paleosalinity (Schouten et al., 2006; Wolhowe et al., 2009; Sachs and Kawka, 2015).

Additionally, the effect of temperature on $\delta^2 H_{C37}$ ratios and $\alpha_{C37}$ values in culture is not well constrained, with some studies



showing an effect (Wolhowe et al., 2009), and others not (Schouten et al., 2006). These extracellular parameters have primarily been investigated for Group III marine haptophytes, with little known about Group II species.

Here, we investigate the relationship between salinity and $\delta^2$H ratios for a variety of haptophyte lipids to explore the impact of

biosynthetic pathways. For this purpose, Group II and Group III haptophytes were grown at different salinities and $\delta^2$H ratios of long-chain alkenones, $C_{14:0} - C_{18:1}$ fatty acids, brassicasterol, and phytol were measured. A secondary aim is to provide the first characterization of $\delta^2H_{C37}$ ratios from the Group II haptophyte *Tisochrysis lutea*, which was grown in batch cultures to assess effects of temperature and nutrient concentrations.

## 2 Methods

### 2.1 Batch cultures of *Emiliania huxleyi*, *Isochrysis galbana* and *Ruttnera lamellosa*

Batch cultures of *E. huxleyi* (CCMP 1516), *I. galbana* (CCMP 1323), and *R. lamellosa* (formerly *Chrysotila lamellosa*, CCMP 1307) were grown over a range of salinities from 26 – 37 for *E. huxleyi*, and 10 – 35 for both *I. galbana* and *R. lamellosa*. Cultures were grown at a temperature of 15° C, and light intensity of 60 µmol photons m$^{-2}$ s$^{-1}$ supplied by cool white fluorescent light with a light:dark cycle of 16:8 h. Specific media conditions are outlined in Chivall et al. (2014 – *R. lamellosa*) and

M'Boule et al. (2014 – *E. huxleyi* and *I. galbana*). Aliquots from the starter culture were transferred to fresh media of each salinity five times prior to the experiment in order to acclimate algae to the new conditions and remove any potential memory effects. Biomass was filtered onto pre-combusted GF/F filters during exponential growth phase when cell concentrations were above 1 x 10$^6$ cells / mL. Filters were freeze-dried, and organics were extracted ultrasonically using dichloromethane / methanol 2:1 (v/v) to obtain total lipid extracts (TLE). TLEs were split in two for *E. huxleyi* and *I. galbana*. One part of the

TLE was separated using an aluminium oxide column into three fractions: apolar (eluted with three column volumes of 9:1 (v/v) hexane / DCM), alkenone (eluted with four column volumes of 1:1 (v/v) hexane / DCM) and polar (eluted with four column volumes of 1:1 (v/v) DCM / MeOH; containing phytol and brassicasterol).

To obtain clean fractions for phytol and brassicasterol analyses, an aliquot of polar fractions from the TLE was saponified using 1N KOH / MeOH (10 mL; 1 h reflux). After cooling, the solvent was neutralised with HCl / MeOH and transferred into

a separating funnel with 3 mL of bidistilled water. Lipids were extracted from the aqueous solution into 3 × 3 mL DCM. The

solvent was removed under vacuum and the extracts dried over $Na_2SO_4$. An aliquot of this saponified polar fraction was cleaned over an aluminium oxide column, eluting with 1:1 (v/v) DCM/MeOH. Phytol and brassicasterol, contained in the saponified polar fraction, were acetylated with 100 µL acetic anhydride ($δ^2H$= -126 ‰) at 60° C for 30 min in the presence of a small of amount of $Ag(CF_3SO_3)$ (Das and Chakraborty, 2011). After cooling, 2 mL of a saturated aqueous solution of $NaHCO_3$ was

added and the acetylated lipids extracted into 3 × 1 mL ethyl acetate. The acetylated lipids were then cleaned over an aluminium oxide column, using 1:1 (v/v) hexane / DCM. Fatty acid extraction was conducted as described by Heinzelmann et al. (2015) for both *E. huxleyi* and *I. galbana*. Fatty acids were not analyzed for *R. lamellosa*.

## 2.2 Batch cultures of *Tisochrysis lutea*

Two strains of *T. lutea,* CCMP463 and NIES-2590, were grown in artificial seawater with added nutrients at a salinity of 30,

light intensity of 100 µmol photons $m^{-2}$ $s^{-1}$ and a range of temperatures from 10 to 35° C. Cells were harvested during late linear and early stationary phases. Alkenones were extracted as described in Nakamura et al. (2016). *T. lutea* strain CCAP 927/14 was grown in triplicate in filtered seawater under N-replete and N-limited conditions with salinity of 34 – 35, temperature from 20-23° C, and light intensity of 180 – 220 µmol photons $m^{-2}$ $s^{-1}$ supplied by cool white fluorescent light. Cells were harvested after 4 and 10 days of growth. Neutral lipid fractions were extracted as described in da Costa et al. (2017),

and ketone fractions were isolated over an aluminum oxide column using 1:1 (v/v) hexane:DCM.

## 2.2 Hydrogen isotope ratios

Hydrogen isotope ratios of growth water were measured prior to starting experiments and after filtration for *E. huxleyi*, *I. galbana*, and *R. lamellosa*. Culture water for the *T. lutea* nutrient experiment was measured prior to starting the experiments only, and no culture water was available for isotope analysis from the *T. lutea* temperature cultures. Water isotope ratios were

measured following methods outlined in Weiss et al. (2017). Biomarkers were quantified using a gas-chromatograph coupled to a flame ionization detector (GC-FID) before measuring hydrogen isotope ratios on a Thermo Scientific Delta V GC/TC/irMS. For biomarkers from *E. huxleyi, I. galbana* and *R. lamellosa,* the GC was equipped with a 25 m CPSil 5 (Agilent, 25 m x 0.32 mm x 0.4 µm) GC column following M'Boule et al. (2014). Due to lack of baseline separation when using a 25 m CPSil 5 GC column, alkenone $δ^2H$ ratios from these three species are the integrated $C_{37:3}$ and $C_{37:2}$ values ($δ^2H_{C37}$). Alkenones

from *T. lutea* were measured by GC/TC/irMS equipped with a RTX-200 60 m column (Restek, 60 m x 0.32 mm x 0.5 μm), allowing for determination of $\delta^2H$ ratios of individual alkenones. The GC temperature program is as follows: 70° C to 250° C at 18° C/min, 250° C to 320° C at 1.5° C/min, then kept at 320° C for 25 min. Helium was used as a carrier gas and the flow rate was 1.5 mL/min. The integrated $\delta^2H_{C37}$ ratios, determined by a second peak integration encompassing both the $C_{37:3}$ and

$C_{37:2}$ alkenone peaks, are also reported for comparison with alkenones from *E. huxleyi, I. galbana* and *R. lamellosa* measured on the CPSil 5 GC column. Prior to running samples each day, the $H_3^+$ factor was measured and corrected for. Values for the $H_3^+$ factor were $2.765 \pm 0.468$ ppm nA$^{-1}$ for analytical runs of *R. lamellosa* lipids, $2.752 \pm 0.428$ ppm nA$^{-1}$ for *I. galbana* lipids, and $4.644 \pm 0.648$ ppm nA$^{-1}$ for *E. huxleyi* lipids and $3.229 \pm 0.261$ ppm nA$^{-1}$ for  *T. lutea* lipids. An n-alkane standard, Mix B, supplied by A. Schimmelmann (Indiana University) was measured, and samples were only run once the average deviation

from the offline determined value and standard deviation for the Mix B were both less than 5 ‰. $H_2$ gas of known isotopic composition was measured at the beginning and end of each analytical run to further monitor machine stability. Squalane and $C_{30}$ n-alkane were co-injected with each run as another control on machine accuracy. Values for squalene were -167 ± 4 ‰ and for $C_{30}$ were -71 ± 4 ‰ for the entire sample set (n = 327). Standard deviations for $\delta^2H$ ratios represent the reproducibility between duplicate or triplicate analytical runs, and generally fall within the 3 ‰ precision window for the Thermo Scientific

Delta V.

**3 Results**

Hydrogen isotope ratios and fractionation of six different lipids were measured and calculated, respectively, from extracts of batch cultures of four different haptophyte species (Supplementary Table 1). The $\delta^2H_{C37}$ ratios from the Group III haptophyte

*E. huxleyi* and the Group II haptophytes *I. galbana* and *R. lamellosa* were previously published by M'Boule et al. (2014) and Chivall et al. (2014) and the $\delta^2H$ ratios of $C_{14:0} - C_{18:1}$ fatty acids for *I. galbana* were previously published by Heinzelmann et al. (2015). Here we report the $\delta^2H_{C37}$ ratios from the Group II haptophyte *T. lutea* cultivated at different salinities, temperature and nutrient conditions. Furthermore, we analyzed the $\delta^2H$ ratios of $C_{14:0} - C_{18:1}$ fatty acids from *E. huxleyi,* and brassicasterol and phytol from *E. huxleyi*, *I. galbana*, and *R. lamellosa*  from the same culture material as that of M'Boule al. (2014) and

Chivall et al. (2014) (Supplementary Table 1). Fractionation factor α values were calculated for all compounds except for alkenones from the *T. lutea* temperature experiment because culture water was no longer available at the time of analyses.

### 3.1 Hydrogen isotope ratios of alkenones from *T. lutea*

#### 3.1.1 Temperature experiment

Group II haptophyte *T. lutea* was grown over a temperature range of 15 – 35° C. Combined $\delta^2H_{C37}$ ratios ranged from -123 ‰ to -152 ‰, while isotope ratios of individual alkenones ranged from -111 ‰ to -153 ‰ for $C_{37:3}$ and -117 ‰ to -160 ‰ for $C_{37:2}$ (Table 1, Fig. 1). Growth was monitored by chlorophyll fluorescence, thus growth rates cannot be easily compared with rates calculated from daily cell counts as done for the other species. However, cells appeared to grow fastest at 35° C and slowest at 15° C (Nakamura et al., 2016). Similarly, alkenone concentrations were stable between 20° C and 30° C, and slightly lower at 35° C and 15° C (Nakamura et al., 2016). The hydrogen isotopic composition of the integrated $C_{37}$ alkenones has a significant positive correlation with temperature ($r = 0.80$, $n = 28$, $p < 0.001$), as does the $C_{37:3}$ ($r = 0.75$, $p < 0.001$), while the $C_{37:2}$ only has a moderate positive correlation with temperature ($r = 0.39$, $p < 0.05$).

#### 3.1.2 Nutrient experiment

To understand the impact of nutrients on hydrogen isotope ratios of alkenones from Group II haptophytes, *T. lutea* was grown under N-replete (nitrate concentration of 1.2 mM) and N-reduced (nitrate concentration of 0.6 mM) conditions. Cells were harvested in exponential and stationary growth phases (after 4 and 10 days of growth respectively). Growth rates for N-replete conditions were 0.21 div. $d^{-1}$, and N-reduced conditions were 0.14 div. $d^{-1}$. Under N-replete conditions, $\delta^2H_{C37}$ ratios ranged from -123 ‰ to -134 ‰, and were more depleted than under N-reduced conditions by around 10 ‰, where $\delta^2H_{C37}$ ratios ranged from -113 ‰ to -121 ‰ (Table 2; Fig. 2a). There is also a clear distinction between growth phases, with more depleted $\delta^2H_{C37}$ ratios corresponding to exponential growth phase (Fig. 2a). Fractionation factor $\alpha_{C37}$ ranged from 0.860 to 0.880, with higher fractionation (lower $\alpha_{C37}$) for higher nutrient conditions (Fig 2b). Higher fractionation was also observed during exponential growth phase compared to stationary growth phase (Fig. 2b). Higher nutrient concentrations led to faster growth, thus a negative correlation between fractionation and growth rate is noted (Fig. 2c). $\delta^2H_{C37}$ ratios from both *T. lutea* experiments align well with other Group II species *I. galbana* and *R. lamellosa*, even when temperature and nutrient concentrations caused a 50 ‰ range in $\delta^2H_{C37}$ ratios (Fig. 3). Hydrogen isotope ratios of individual $C_{37}$ alkenones follow the same trends as the

integrated $C_{37}$ alkenones (Table 2). Integrated ratios are used for comparison with $\delta^2H$ ratios of *I. galbana* and *R. lamellosa*

measured on a shorter GC column which prohibited baseline separation of the individual alkenones.

### 3.1 Hydrogen isotope ratios of lipids from *E. huxleyi, I. galbana* and *R. lamellosa*

For *I. galbana* and *R. lamellosa*, alkenones were the most enriched lipids by an average of 60 ‰ and 110 ‰, respectively,

compared to the other lipids (Fig. 4a,b). In *E. huxleyi*, $\delta^2H$ ratios of $C_{14:0}$ and $C_{16:0}$ fatty acids were more depleted by 10 to 30

‰ relative to alkenones, while the $C_{18:1}$ fatty acid was the most $^2H$-enriched compound by around 50 ‰ relative to the other

fatty acids and alkenones (Fig. 4c). Fractionation for *E. huxleyi* fatty acids varied between 0.762 – 0.872, and correlates

significantly with salinity for $C_{14:0}$ and $C_{18:1}$ fatty acids (Fig. 4c), but the correlation between fractionation and salinity is not

statistically significant for $C_{16:0}$ ($r = 0.84$, $n = 4$, $p = 0.16$). However, we treat these significant correlations with caution as they

are based on only four data points. The $\delta^2H$ ratios of both brassicasterol and phytol are more depleted than $\delta^2H$ ratios of

alkenones by an average of 120 ‰ and 230 ‰, respectively, for all three species (Fig. 4; Supplementary Table 1). The $\delta^2H$

ratios and $\alpha$ of brassicasterol correlate significantly with salinity in all three species (Fig. 4). There is no significant relationship

with salinity for phytol from *E. huxleyi* and *I. galbana*, but both $\delta^2H_{phytol}$ ratios and $\alpha_{phytol}$ from *R. lamellosa* do correlate

significantly with salinity ($\delta^2H_{phytol}$: $r = 0.90$, $n = 28$, $p < 0.001$; $\alpha_{phytol}$: $r = 0.54$, $n = 28$, $p < 0.05$).

## 4 Discussion

### 4.1 Effects of temperature for *T. lutea* alkenones

The influence of temperature on $\delta^2H_{C37}$ ratios in haptophyte algae is not well constrained. Previous results from cultures

investigating the effect of temperature on $\delta^2H_{C37}$ ratios in Group III haptophyte species are contradictory (Schouten et al.,

2006; Wolhowe et al., 2009), and there has been no characterization of the temperature effect on $\delta^2H_{C37}$ ratios for Group II

haptophytes to our knowledge. Our results show that individual alkenones, as well as integrated $C_{37}$ alkenones, show a

significant positive correlation with temperature in Group II species *T. lutea* (Fig. 1). The positive linear correlation for the

integrated alkenones seems to be driven by the $C_{37:3}$ alkenone, begging the question whether the observed correlation is really

a temperature effect, or primarily a function of relative abundance and therefore an indirect temperature effect (c.f. van der

Meer et al., 2013).

The relative abundance of the $C_{37:3}$ and $C_{37:2}$ alkenones are strongly coupled to temperature, with higher amounts of the $C_{37:2}$ at higher temperatures (Prahl and Wakeham, 1987). The $C_{37:3}$ is synthesized from the $C_{37:2}$ by a desaturation step (Rontani et al., 2006; Endo et al., 2018; Kitamura et al., 2018), which occurs less often at higher temperatures. Desaturation is associated

with $^2H$ depletion (Chikaraishi et al., 2004), thus it follows that the $C_{37:3}$ alkenone should be more depleted than its $C_{37:2}$ precursor, as observed here (Fig. 1). At lower temperatures, when there is a higher abundance of $C_{37:3}$, and much lower relative abundance of $C_{37:2}$, the $C_{37:2}$ should be comparatively enriched. For the *T. lutea* temperature experiment, the $C_{37:3}$ was depleted relative to the $C_{37:2}$ from 15 – 25° C, but from 30 – 35° C, where the $C_{37:2}$ was much more abundant than the $C_{37:3}$, a relative depletion of the $C_{37:2}$ compared to the $C_{37:3}$ was observed (Fig. 1). It appears that $^2H$ depletion shifts in favor of the dominant

alkenone, and when the difference in abundance of both alkenones is close to zero, the offset between the $\delta^2H$ ratios of the two alkenones is also close to zero. These findings suggest that the temperature correlation observed for alkenones from *T. lutea* is likely related to relative abundance shifts, which in turn has an effect on $\delta^2H_{C37}$ ratios, implying an indirect temperature effect on $\delta^2H_{C37}$ ratios. Growth rate might also influence these correlations, but due to the fact that growth was monitored by chlorophyll fluorescence in these experiments, we cannot compare absolute values with growth rates obtained by daily cell

counts. However, alkenone concentrations did not vary substantially over the temperature range (5-10 µg / mL at 15° C and 35° C vs 10-15 µg / mL from 20 -30° C; Fig. 2b from Nakamura et al., 2016), leading us to conclude that growth rates did not significantly impact our $\delta^2H_{C37}$ ratios.

### 4.2 Effects of nutrients and growth phase for *T. lutea* alkenones

*T. lutea* was also grown under two different nutrient concentrations and harvested during exponential and stationary growth phases to assess growth-related effects on $\alpha_{C37}$ values. N-reduced and N-replete media led to different growth rates: 0.14 µ d$^{-1}$ for N-reduced and 0.21 µ d$^{-1}$ for N-replete. While these two growth rates are not extremely different, they led to a distinct offset in isotope ratios, with more fractionation under higher nutrient concentrations (i.e. higher growth rate) and a relative isotope depletion (Fig. 2). The negative correlation between $\alpha_{C37}$ values and growth rate was previously recognized in Group

III species *E. huxleyi* and *G. oceanica* (Schouten et al., 2006; Wolhowe et al., 2009). The same negative relationship between $\alpha_{C37}$ values and growth rate was noted for our *T. lutea* experiment (Fig. 2), confirming that similar growth effects on $\delta^2H_{C37}$

ratios also occur in Group II species and are tightly coupled to nutrient availability. Sachs and Kawka (2015) hypothesized

that lower growth rate leads to an up-regulation of OPP derived NADPH causing lipids to be relatively enriched; at higher

growth rates, photosynthetically derived NADPH becomes the dominant H source, leading to a relative depletion of lipid $\delta^2$H

ratios. This mechanism works in reverse to the salinity mechanism, where increased salinity causes an up-regulation of OPP

5 derived NADPH, and therefore might account for the reduced sensitivity of $\delta^2H_{C37}$-salinity in the natural environment (e.g.

Schwab and Sachs, 2011; Weiss et al., 2019). The $\delta^2H_{C37}$ ratios from *T. lutea* (temperature and nutrient experiments) fit well

with values noted for other Group II species *I. galbana* and *R. lamellosa* (Fig.3). Three Group II species now show similar

$\delta^2H_{C37}$ ratios which are more enriched than values reported for the Group III species *E. huxleyi* and *G. oceanica*, corroborating

the hypothesis of a distinct Group II and Group III $\delta^2H_{C37}$ signal (M'Boule et al., 2014).

### 4.3 Impact of salinity of haptophyte lipids
#### 4.3.1 Fatty acids and long-chain alkenones

Fatty acids from *E. huxleyi* in our experiment showed $^2$H-enrichment with chain elongation (Supplementary Table 1, Fig. 4).

The same has previously been observed for fatty acids from *E. huxleyi* (Sachs and Kawka, 2015), and *I. galbana* (Heinzelmann

15 et al., 2015), for the diatom *Thalassiosira pseudonana* (Maloney et al., 2016), as well as in some marine macroalgae belonging

to Heterokontophyta and Rhodophyta (Chikaraishi et al., 2004). Subsequent depletion is reported after desaturation from $C_{18:0}$

to $C_{18:1}$ of around 40 ‰ for Heterokontophyta and approximately 70 ‰ for Rhodophyta, but the overall combined effect of

elongation and desaturation from $C_{16:0}$ to $C_{18:1}$ results in isotopic enrichment of $C_{18:1}$ relative to $C_{16:0}$ (Chikaraishi et al., 2004).

In general, these observations also hold true for fatty acids from *E. huxleyi* and *I. galbana* discussed here (Supplementary

20 Table 1, Fig, 4). Both species show a significant positive correlation between $\delta^2$H ratios of fatty acids and salinity (Fig. 4),

except for the α - salinity correlation for $C_{16:0}$ from *E. huxleyi* and $C_{18:1}$ from *I. galbana* ($C_{16:0}$ : r = 0.84, p > 0.05; $C_{18:1}$ : r = -

0.34, p > 0.05). However, the correlations for fatty acids from *E. huxleyi* are based on a low number of data points (n = 4 for

each fatty acid), thus we treat these significant correlations with caution. The lack of correlation for $C_{18:1}$ in *I. galbana* might

partially be explained by the desaturation step since desaturation is associated with $^2$H depletion, which has the potential to

25 counteract the $^2$H enrichment associated with higher salinities.

The $\delta^2H$ ratios and $\alpha$ vales of alkenones from *I. galbana* (M'Boule et al., 2014), *R. lamellosa* (Chivall et al., 2014)*, E. huxleyi* (M'Boule et al., 2014) all show strong positive correlations with salinity (Supplementary Table 1, Fig. 4). Interestingly, for *E. huxleyi*, the $C_{18:1}$ is the most $^2H$-enriched compound out of the six, in contrast to *I. galbana*, where alkenones are the most $^2H$-enriched (Supplementary Table 1, Fig. 4). This highlights another potential offset between haptophyte Groups II and III.

Species related differences aside, there is a correlation between $\delta^2H$ ratios and salinity for both fatty acids and alkenones. In algae, the first two steps of fatty acid biosynthesis (formation of malonyl-CoA and initial elongation by fatty acid synthases) occur in the chloroplast, and additional elongation beyond $C_{16}$ or $C_{18}$ relies directly on cytosolic sources (Cook and McMaster, 2002; Huerlimann and Heimann, 2013). Fatty acid elongation occurs via the stepwise addition of two carbon atoms and four hydrogen atoms, and desaturation removes two hydrogen atoms, allowing for substantial isotopic change (Chikaraishi et al.,

2004), which can explain the isotopic differences between fatty acids and alkenones. Furthermore, alkenones are thought to be synthesized from these shorter chain fatty acids by elongation and subsequent decarboxylation in the chloroplast (Rontani et al., 2006). Fatty acids, although mainly synthesized in the chloroplast, also rely on acetyl-CoA as a precursor. Acetyl-CoA might be responsible for the salinity signal noted for fatty acids and alkenones, since *de novo* synthesis of acetyl-CoA comes from pyruvate produced in the cytosol (DeNiro and Epstein, 1977).

### 4.3.2 Brassicasterol and phytol

The $\delta^2H$ ratios and $\alpha$ values of brassicasterol correlate significantly with salinity for all three species, whereas $\delta^2H$ ratios of phytol do not correlate with salinity for either *E. huxleyi* or *I. galbana* (Fig. 4). Brassicasterol and phytol are isoprenoid lipids formed with either isopentenyl phosphate (IPP) or dimethylallyl pyrophosphate (DMAPP) as precursors. Two pathways exist

for the synthesis of IPP and DMAPP. The Mevalonate (MVA) Pathway generates isoprenoid precursors in the cytosol using acetyl-CoA (Schmidt et al., 2003; Eisenreich et al., 2004). In contrast, the Methylerythritol pathway (MEP) creates IPP and DMAPP in the plastids using pyruvate and glyceraldehyde-3-phosphate (G3P) (Schmidt et al., 2003; Guggisberg et al., 2014; Lohr et al., 2012; Sachs et al., 2016). Since phytol is synthesized by precursors generated via MEP and brassicasterol is synthesized by precursors from MVA (Lichtenthaler et al., 1997; Bohlmann et al., 1999; Lichtenthaler, 1999; Chikaraishi et

al., 2009), the offset of ~100 ‰ between the two compounds likely stems from differences associated with these two pathways. Synthesis of isoprenoid precursors via the MVA pathway involves two reduction steps, but the MEP involves three reduction

steps. One of the three reduction steps in the MEP is catalyzed by the IspH enzyme that supplies IPP and/or DMAPP with

depleted H (Schmidt et al., 2003), potentially explaining why phytol is more $^2$H depleted than brassicasterol.

Further evidence to confirm different precursor pathways and cellular compartments for phytol versus sterol comes from a

light intensity experiment conducted by Sachs et al. (2017) assessing effects of irradiance on organic compounds from the

diatom *Thalassiosira pseudonana.* The $\delta^2$H ratios of phytol showed a strong correlation with light intensity ($R^2 = 0.90$, p <

0.0001), while the $\delta^2$H ratio of 24-methyl-cholesta-5,24(28)-dien-3$\beta$-ol was not affected by light intensity. These results

support the idea that phytol is synthesized by precursors in the chloroplast, especially since phytol is a component of

chlorophyll (de Souza and Nes, 1969). Sterols, on the other hand, can be synthesized by cytosolic precursors and rely on a H

source that is not connected to light availability, but instead potentially correlated to salinity, either directly or indirectly.

Acetyl-CoA used in the MVA pathway is synthesized from pyruvate or $\beta$-oxidation of excess fatty acids in the cytosol (Nelson

and Cox, 2017). Pyruvate can be synthesized in the chloroplast, and used directly in MEP, but it can also be synthesized in the

cytosol, where it can be converted to acetyl-CoA and used in MVA (Williams and Randall, 1979; Disch et al., 1998; Wallace,

2013). Differences in cellular compartment and biosynthetic precursors (acetyl-CoA and pyruvate) associated with metabolic

reactions provide an explanation for why $\delta^2$H ratios of brassicasterol correlate to salinity and $\delta^2$H ratios of phytol do not.

However, in *R. lamellosa,* there is a positive linear correlation between phytol $\delta^2$H and salinity ($\delta^2$H: r = 0.90, n= 18,  p <

0.001), and a much weaker correlation for $\alpha$ (r = 0.54, n = 18,  p < 0.05). The correlation with salinity observed for phytol

from *R. lamellosa* is puzzling. It might be related to an addition of pyruvate synthesized in the cytosol mixing with pyruvate

synthesized in the chloroplast, since pyruvate is reported to flow between the two cellular compartments (Williams and

Randall, 1979).

### 4.4 Mechanisms for salinity impact on hydrogen isotope fractionation

The general trend observed between $\delta^2$H ratios of organic compounds (excluding phytol) and salinity is $\delta^2$H enrichment with

increasing salinity. As explained above, the absence of a correlation between $\delta^2$H ratios of phytol and salinity is presumably

the result of the location of phytol biosynthesis and biosynthesis of the key intermediate, pyruvate. Fatty acids, alkenones and

brassicasterol are all connected more directly to H pools in the cytosol because of their reliance on acetyl-CoA as a precursor, and all show relative $^2$H-enrichment at increased salinities (Fig. 4). While the precise mechanism controlling the correlation between $\delta^2$H ratios and salinity is unknown, certain concepts related to hydrogen isotope fractionation and intracellular pools of hydrogen are known.

NADPH is an essential cofactor that acts as a significant source of H in biosynthesis reactions (Sessions et al., 1999; Zhang et al., 2009 and references cited therein). The hydrogen isotopic composition of NADPH can vary substantially depending on the source or location of reduction within the cell, which can then be evidenced down-stream in the hydrogen isotope ratios of resultant organic compounds (Zhang et al., 2009). NADPH can be reduced via two main sources: light reactions in the chloroplast (photosynthetically-derived) and the oxidative pentose phosphate (OPP) pathway in the cytosol (metabolically-derived). Photosynthetically-derived NADPH is the product of the reduction of NADP+ by the redox protein ferredoxin (Shin, 2004; Zhang et al., 2009; Cormier et al., 2018). The resultant NADPH is $^2$H-depleted compared to growth water. The $\delta^2$H ratios of NADPH reduced by light reactions of photosynthesis are dependent on light intensity below ~ 115 $\mu$ mol photons m$^{-2}$ s$^{-1}$ and large fractionation is associated with lower light intensities (Cormier et al., 2018). In the cytosol, the OPP pathway supplies the cell with NADPH by reducing NADP+ during conversion of glucose-6-phosphate (Wamelink et al., 2008), a reaction that is not linked to photosynthetic activity, but potentially linked to salinity (Sachs et al., 2016). It has been speculated that at lower light intensities, the OPP pathway supplies a larger portion of NADPH for biosynthesis since NADPH from light reactions of photosynthesis is limited (Cormier et al., 2018). Additionally, up-regulation of the OPP pathway is hypothesized at increased salinities, which is associated with relative $^2$H-enrichment (Maloney et al., 2016; Sachs et al., 2016). A dominance of metabolically-derived H for biosynthesis would yield more enriched organic compounds relative to a when a photosynthetically-derived H source is dominant, and this enrichment is enhanced at increased salinities. Isotopic enrichment is observed at increased salinities in a majority of the lipids discussed here (Fig. 4), leading to the hypothesis that OPP is likely the dominant H source of NADPH used for synthesis of these lipids. For haptophytes not growing under light-limited conditions, but growing over a range of salinities, a balance between photosynthetically reduced and OPP derived NADPH is expected at lower salinities, and a larger proportion of OPP derived NADPH is predicted at higher salinities. Phytol synthesis



occurs in the chloroplast, thus a significant portion of H should come from photosynthetically-derived NADPH, explaining why phytol $\delta^2H$ ratios are significantly depleted relative to other organic compounds, and $\delta^2H$ ratios of phytol might be more sensitive to light rather than salinity.

The amount of H derived from water for biosynthesis is an equally important control on why some lipids retain a correlation to salinity. For example, when pyruvate is formed by the tricarboxylic acid (TCA) cycle, there is more exchange of H with relatively enriched surrounding water, which causes a relative isotopic enrichment (Sachs et al., 2016; Cormier et al., 2018). In contrast, when pyruvate is formed by glycolysis, only one third of the H comes from water, resulting in a relative depletion (Sachs et al., 2016). However, as explained above, NADPH sources are thought to exert a strong control on $^2H$ ratios of lipids,

thus these differences between pyruvate from TCA compared to glycolysis might be of secondary importance in resultant lipid $^2H$ ratios. When larger amounts of H derived directly from water is used in lipid synthesis, there is a more direct connection between lipid $\delta^2H$ ratios and growth water $\delta^2H$ ratios, and therefore less fractionation. Less fractionation is observed at higher salinities for the majority of biomarkers presented here, therefore it is possible that up-regulation of certain pathways incorporates proportionally more H directly from intracellular water, or potentially employing more acetyl-CoA from TCA-

derived pyruvate, at higher salinities causes a closer connection to extracellular $\delta^2H_{H2O}$ ratios.

At higher salinities cells must maintain osmotic pressure to avoid lysis, and create organic compounds known as osmolytes for this purpose (Dickson et al., 1982; Dickson and Kirst, 1987). Indeed, higher amounts of osmolyes (most notably DMSP) are observed at higher salinities in some haptophyte species (Dickson and Kirst, 1987). It has been previously proposed that

enhanced production of osmolytes might cause the intracellular pool of H to become more $^2H$ enriched as a result of increased recycling of intracellular water (Sachse and Sachs, 2008; Maloney et al., 2016), leading to enriched $\delta^2H$ ratios of lipids at higher salinities. *I. galbana* is adapted to grow at a large range of salinities, while *E. huxleyi* is adapted to grow at a higher, but more restricted range of salinities. Because *I. galbana* is more cosmopolitan, it might be more metabolically active as a consequence of salinity stress over the much larger range of salinities at which it grows, and could likely be shifting more

precursors and reducing power into osmolyte production than *E. huxleyi* at the same salinity. This would cause relative

enrichment of the intracellular H pool in *I. galbana* compared to *E. huxleyi*, explaining offset in $\delta^2$H ratios observed between these two species. The enrichment associated with increased osmolyte production might not be as strongly evidenced in fatty acids because fatty acids are initially synthesized in the chloroplast, up to $C_{14}$ or $C_{16}$, thus are not exposed directly to cytosolic pools of H. Chain elongation leading to long-chain alkenones does take place in the cytosol, thus $\delta^2$H ratios of alkenones

reflect changes in salinity.

The above mechanisms suggest that the site of NADPH reduction is the primary driver of the hydrogen isotope composition of lipids, but the source of the salinity signal recorded in haptophyte lipids potentially comes from biosynthetic precursors tied to specific cellular compartments. The hydrogen isotopic composition of compounds formed from cytosol-derived precursors,

and those synthesized directly in the cytosol, change with salinity. Lipids that are synthesized in the chloroplast and depend more on chloroplastic precursors in conjunction with photosynthetically-derived NADPH, do not show variation in $\delta^2$H ratios with salinity (e.g. phytol in *E. huxleyi* and *I. galbana*), or greatly diminished (e.g. $C_{16:0}$ in *E. huxleyi*).

**5 Conclusions**

New results presented here showed a positive effect of temperature on $\delta^2$H ratios of long-chain alkenones for Group II species *T. lutea* which is likely coupled with changes in relative abundance of individual alkenones. Nutrient concentration showed a similar effect on Group II species *T. lutea* as previously reported for Group III species *E. huxleyi,* with depletion of $\delta^2$H ratios associated with stationary growth phase and higher nutrient concentrations. The effect of nutrients and growth on $\delta^2$H ratios noted here is opposite to the effect of salinity, and might cause changes in the sensitivity of the $\delta^2$H-salinity relationship in the

natural environment.

Investigation of $\delta^2$H ratios of multiple lipids produced by the same haptophyte algae showed that biosynthesis occurring in the cytosol, and relying on precursors of cytosolic origin, yields lipids with $\delta^2$H ratios that are sensitive to salinity. In contrast, the $\delta^2$H of lipids produced in the chloroplast, and dependent upon precursors partially or whole produced in the chloroplast, did

not change with salinity. Therefore, alkenones, brassicasterol and fatty acids with a chain length of at least 16 carbon atoms,

or diagenetic products derived from these lipids, should be selected instead of phytol or shorter chain fatty acids for paleosalinity reconstructions. The offset between $\delta^2$H ratios of lipids synthesized by Group II and III haptophytes is likely a result of differing mechanisms for regulating salt stress. Group III haptophytes are plausibly more adapted to growth at higher salinities than Group II haptophytes, which are adapted for growth at a more variable range of salinities. The offset between

species, which can be on the order of 100 ‰, is an essential factor to consider when employing $\delta^2$H ratios to reconstruct salinity in areas where mixing of species is anticipated.

**Author Contributions**

D. C and S. K. designed and carried out the *E. huxleyi, I. galbana* and *R. lamellosa* experiments and the respective hydrogen
isotope analyses. H. N. provided the extracts for the *T. lutea* temperature experiment. F. D. C. and P. S. provided the extracts for the *T. lutea* nutrient and growth phase experiment. G. M. W. conducted hydrogen isotope analyses for both *T. lutea* experiments and prepared the manuscript with contributions from all co-authors.

**Acknowledgements**

We would like to thank both Anna Noordeloos and Hiroya Araie for help with culturing. Dr. D.X. Sahonero-Canavesi for extensive discussions about intracellular mechanisms. This study received funding from the Netherlands Earth System Science Center (NESSC) though a Gravitation grant (024.002.001) from the Dutch Ministry for Education, Culture and Science. This research was also supported by the Core Research for Evolutional Science and Technology in Japan Science and Technology Agency (CREST/JST), Japan. All acquired data will be stored in the Pangaea database.

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



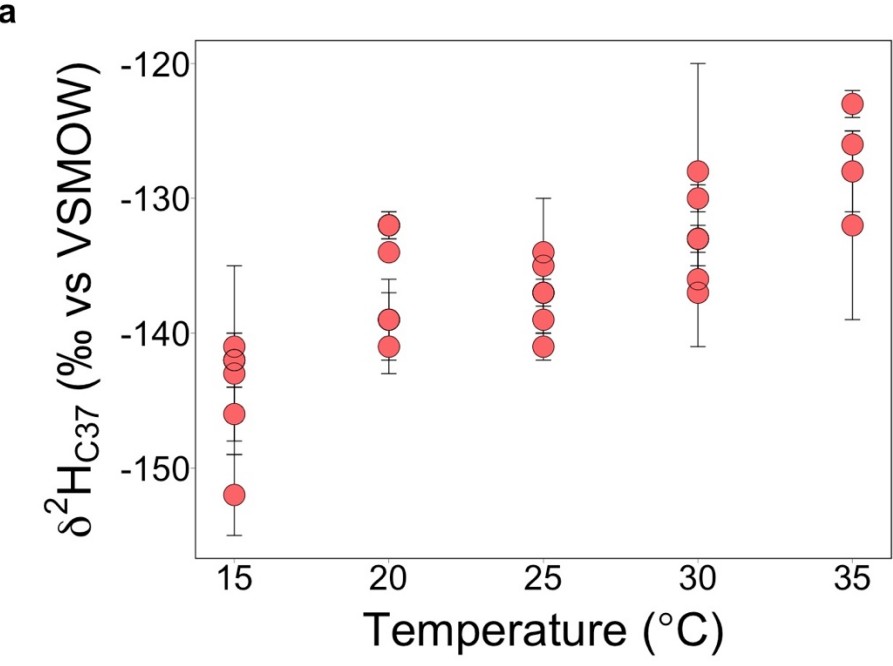

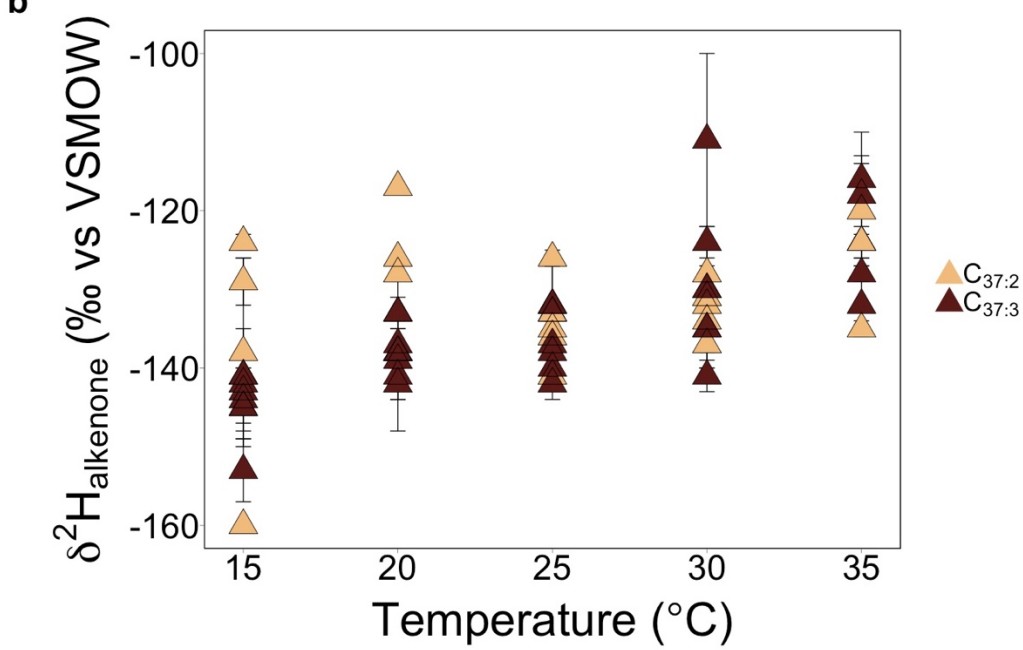

**Figure 1: Hydrogen isotope ratios of (a) combined and (b) individual C$_{37}$ alkenones ($\delta^2$H$_{C37}$) from Group II species *Tisochrysis lutea* plotted against temperature. Both integrated and individual $\delta^2$H ratios show a positive linear correlation to temperature. Integrated: r = 0.80, p < 0.001; C$_{37:3}$: r = 0.75, p < 0.001; C$_{37:2}$: r = 0.39, p < 0.05; n = 28.**



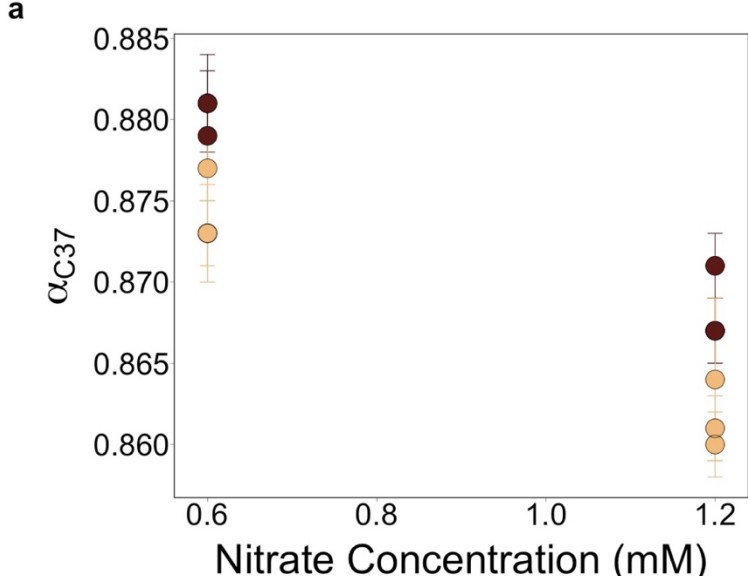

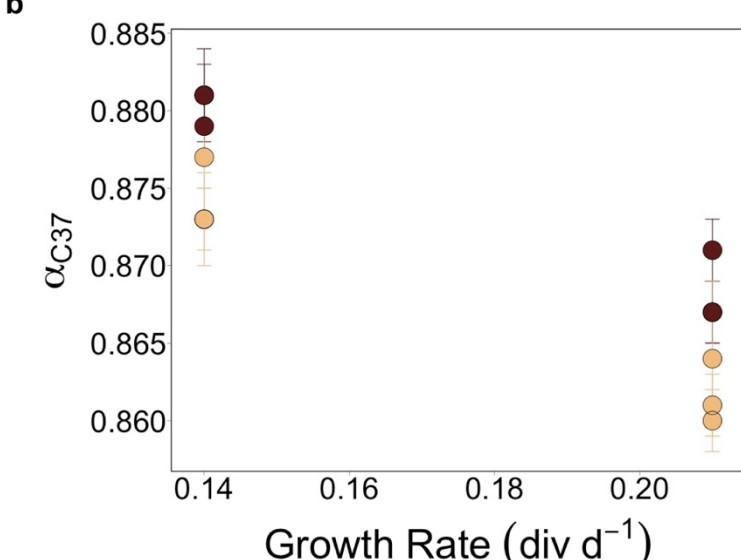

**Figure 2: Results from batch cultures of *Tisochrysis lutea* investigating effect of nutrient concentrations, growth phase and growth rate on hydrogen isotope fractionation of long-chain alkenones ($\alpha_{C37}$). Hydrogen isotope fractionation factor $\alpha_{C37}$ shows (a) a negative trend with nutrient concentration, and (b) a negative trend with growth rate. Relative isotopic enrichment is noted for stationary growth phase compared to exponential growth phase.**





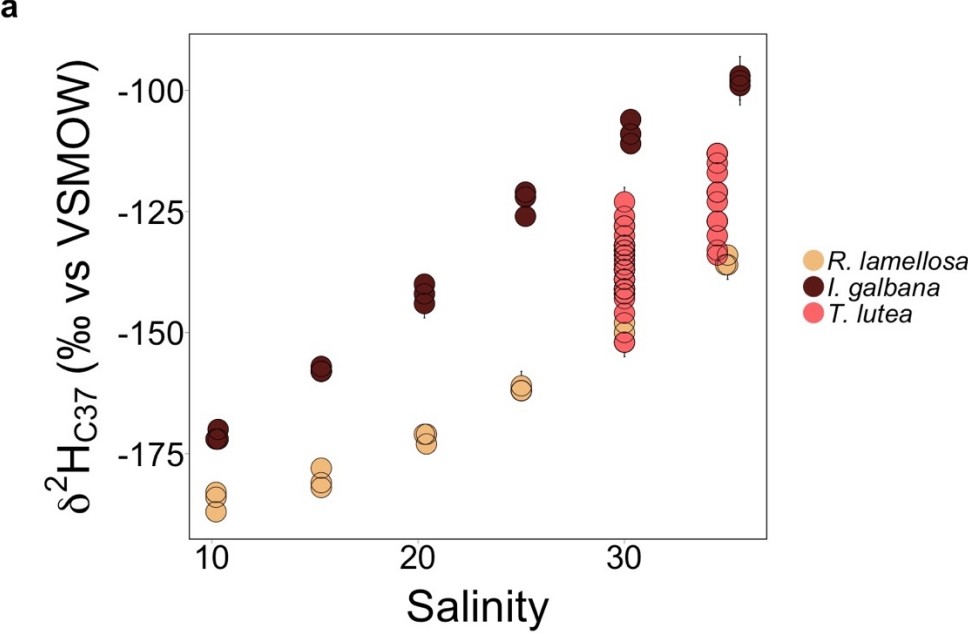

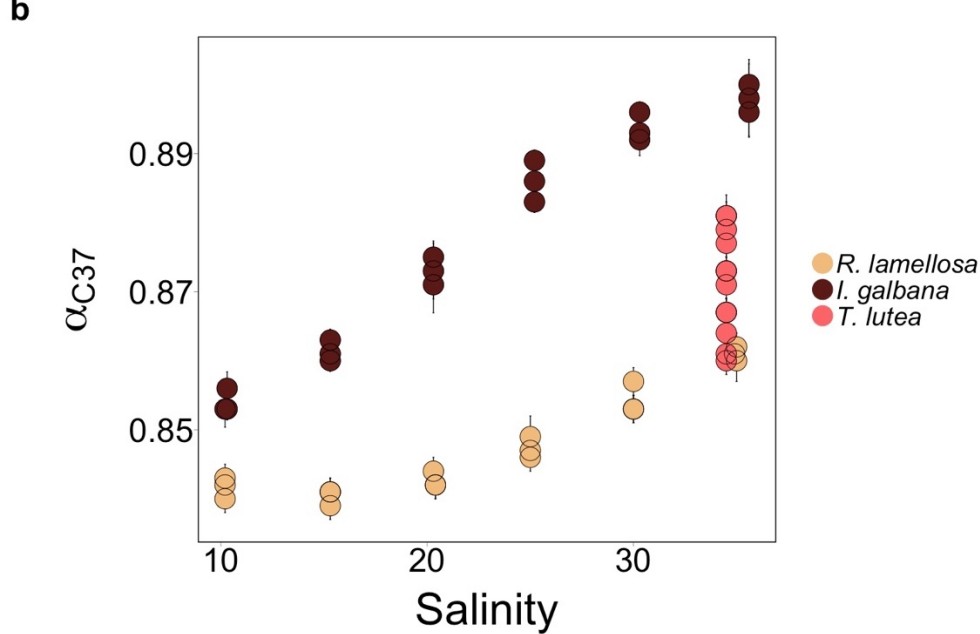

**Figure 3: (a) Hydrogen isotope ratios of long-chain alkenones ($\delta^2H_{C37}$) and (b) hydrogen isotope fractionation factor ($\alpha_{C37}$) plotted against salinity for three Group II species: *Ruttnera lamellosa* (Chivall et al., 2014), *Isochrysis galbana* (M'Boule et al., 2014), and *Tisochrysis lutea* (This study). New results presented here of $\delta^2H_{C37}$ ratios and $\alpha_{C37}$ values from *T. lutea* align with the other two**
5  **species, confirming a distinct Group II signal.**





**Figure 4: Hydrogen isotope ratios ($\delta^2$H) of six biomarker lipids plotted against salinity for Group II species *Isochrysis galbana* (a) and *Ruttnera lamellosa* (b) and Group III species *Emiliania huxleyi* (c). Long-chain alkenones and C14:0 – C18:1 fatty acids from *I. galbana* published by M'Boule et al. (2014) and Heinzelmann et al. (2015) respectively. Long-chain alkenones from *R. lamellosa* and *E. huxleyi* published in Chivall et al. (2014) and M'Boule et al. (2014) respectively. Positive correlations between $\delta^2$H ratios and salinity are noted for all lipids except for phytol synthesized by *I. galbana* and *E. huxleyi*.**





### *Tisochrysis lutea* -- Temperature Experiment

| Strain | Replicate | Temperature (°C) | $\delta^2 H_{C37:3}$ (‰ vs VSMOW) | $\delta^2 H_{C37:2}$ (‰ vs VSMOW) | $\delta^2 H_{C37}$ (‰ vs VSMOW) |
|---|---|---|---|---|---|
| CCMP 463 | A | 15 | -141 ± 6 | -138 ± 12 | -142 ± 7 |
|  | B |  | -144 ± 4 | -141 ± 3 | -143 ± 3 |
|  | C |  | -153 ± 4 | -145 ± 4 | -152 ± 3 |
|  | A | 20 | -138 ± 1 | -138 ± 10 | -139 ± 3 |
|  | B |  | -133 ± 0 | -117 ± 0 | -132 ± 1 |
|  | C |  | -141 ± 3 | -133 ± 2 | -139 ± 2 |
|  | A | 25 | -137 ± 3 | -136 ± 2 | -137 ± 3 |
|  | B |  | -142 ± 2 | -133 ± 1 | -139 ± 2 |
|  | C |  | -132 ± 5 | -135 ± 1 | -135 ± 1 |
|  | A | 30 | -141 ± 2 | -132 ± 1 | -133 ± 0 |
|  | B |  | -111 ± 11 | -130 ± 6 | -128 ± 8 |
|  | C |  | -130 ± 2 | -134 ± 2 | -133 ± 1 |
|  | A | 35 | -118 ± 8 | -124 ± 1 | -123 ± 1 |
|  | B |  | -128 ± 1 | -124 ± 2 | -126 ± 0 |
|  | C |  | -116 ± 2 | -135 ± 1 | -132 ± 7 |
|  | D |  | -132 ± 3 | -120 ± 7 | -128 ± 3 |
|  |  |  |  |  |  |
| NIES 2590 | A | 15 | -143 ± 1 | -129 ± 3 | -142 ± 2 |
|  | B |  | -142 ± 2 | -160 ± 0 | -146 ± 2 |
|  | C |  | -145 ± 0 | -124 ± 1 | -141 ± 0 |
|  | A | 20 | -137 ± 3 | -128 ± 0 | -132 ± 1 |
|  | B |  | -142 ± 2 | -138 ± 3 | -141 ± 2 |
|  | C |  | -139 ± 2 | -126 ± 0 | -134 ± 0 |
|  | A | 25 | -132 ± 5 | -126 ± 1 | -134 ± 4 |
|  | B |  | -138 ± 2 | -133 ± 6 | -137 ± 3 |
|  | C |  | -140 ± 0 | -141 ± 2 | -141 ± 1 |
|  | A | 30 | -124 ± 2 | -128 ± 1 | -130 ± 1 |
|  | B |  | -135 ± 3 | -131 ± 3 | -136 ± 1 |
|  | C |  | -135 ± 5 | -137 ± 3 | -137 ± 4 |

**Table 1: Hydrogen isotope ratios of long-chain alkenones for the *Tisochrysis lutea* temperature experiment. Alkenone concentrations are reported in Nakamura et al. (2016).**





*Tisochrysis lutea* CCAP 927/14 -- Nutrient Experiment

| Replicate | Nitrate mM | $\delta^2H_{H2O}$ (‰ vs VSMOW) | Growth Rate (div d⁻¹) | Phase | $\delta^2H_{C37:3}$ (‰ vs VSMOW) | $\delta^2H_{C37:2}$ (‰ vs VSMOW) | $\delta^2H_{C37}$ (‰ vs VSMOW) | $\alpha_{C37:3}$ | $\alpha_{C37:2}$ | $\alpha_{C37}$ |
|---|---|---|---|---|---|---|---|---|---|---|
| A | 0.6 | 7 ± 2 | 0.14 | Exponential | -121 ± 3 | -119 ± 2 | -121 ± 2 | 0.873 ± 0.003 | 0.875 ± 0.002 | 0.873 ± 0.003 |
| B | | | | | -119 ± 0 | -122 ± 1 | -121 ± 0 | 0.881 ± 0.000 | 0.878 ± 0.001 | 0.879 ± 0.000 |
| C | | | | | -117 ± 1 | -117 ± 0 | -117 ± 1 | 0.883 ± 0.001 | 0.883 ± 0.000 | 0.883 ± 0.001 |
| A | 1.2 | 7 ± 2 | 0.21 | | -135 ±1 | -129 ± 4 | -134 ± 1 | 0.865 ± 0.001 | 0.871 ± 0.005 | 0.866 ± 0.001 |
| B | | | | | -132 ± 1 | -130 ± 1 | -133 ± 1 | 0.868 ± 0.002 | 0.870 ± 0.001 | 0.867 ± 0.002 |
| C | | | | | -130 ± 6 | -128 ± 3 | -130 ± 4 | 0.870 ± 0.007 | 0.872 ± 0.004 | 0.870 ± 0.005 |
| A | 0.6 | 7 ± 2 | | Stationary | -116 ± 1 | -110 ± 0 | -113 ± 0 | 0.884 ± 0.001 | 0.890 ± 0.000 | 0.887 ± 0.000 |
| B | | | | | -118 ± 0 | -110 ± 2 | -115 ± 1 | 0.882 ± 0.000 | 0.890 ± 0.002 | 0.885 ± 0.001 |
| C | | | | | -117 ± 2 | -110 ± 2 | -113 ± 2 | 0.883 ± 0.002 | 0.890 ± 0.003 | 0.887 ± 0.002 |
| A | 1.2 | 7 ± 2 | | | -125 ± 2 | -118 ± 1 | -123 ± 1 | 0.875 ± 0.002 | 0.882 ± 0.001 | 0.877 ± 0.001 |
| B | | | | | -128 ± 1 | -124 ± 5 | -127 ± 0 | 0.872 ± 0.001 | 0.876 ± 0.005 | 0.873 ± 0.000 |
| C | | | | | -129 ± 0 | -120 ± 3 | -127 ± 1 | 0.871 ± 0.000 | 0.880 ± 0.003 | 0.873 ± 0.001 |

**Table 2: Hydrogen isotope ratios ($\delta^2H_{C37}$) and fractionation factor $\alpha_{C37}$ values for the *Tisochrysis lutea* nutrient and growth phase experiment. Alkenone concentrations are reported in da Costa et al. (2017).**