# Peer review of "Impact of metabolic pathways and salinity on the hydrogen isotope ratios of haptophyte lipids"

_Biogeosciences, 2019_

## Referee Comment (RC1) · Anonymous Referee #1 · 27 May 2019

The manuscript by Weiss et al. reports new isotopic compositional data for alkenones, fatty acids, a sterol, and phytol for three different alkenone producers grown under S- and nutrient- experimental conditions. The primary novel contributions the data makes are the interesting responses of alkenone dD to both T and growth phase, in opposition to what has been reported in the literature for other alkenone-producers. They also further document the "salinity effect" on hydrogen isotope fractionation in a number of lipids produced under varying experimental conditions. While the new data is well worth reporting, and may tell very interesting stories about the potential mechanisms behind the temperature, growth rate/phase, and salinity effects, the manuscript as it currently stands has several flaws in its arguments. The data quality, and overall writing,

are worthy of publication - the well-documented experimental section is particularly appreciated. However, major revisions to the overall argumentative thread of the paper will be necessary before this can be vetted.

In no particular order, here are some concerns with the paper that need to be addressed:

1) We really need the T. lutea growth rates for the temp experiments. It is true that, depending on culture density, you may have divergence of chlorophyll fluo. growth rate from cell count growth rate, due to shading, but it is still better than nothing. Given that the temperature effect is completely the opposite of what has been seen before (note that in addition to the somewhat-indeterminate alkenone work, the negatively-sloped temperature effect is also seen in other lipids by Zhang et al. 2009, Organic Geochem), and given the growth rate effects shown here and elsewhere, an attempt should be made at least to constrain how much temp-dependent rate change may impact (counteract?) this curve. The note about the approximately-identical per-volume alkenone concentrations is potentially useful, but only if the cultures were all inoculated at exactly the same density, took off identically with identical lag phases, etc. F curves would be more useful.

2) We also need to see growth curves for the nutrient experiment (f-based or otherwise). "Day 4 and day 10" doesn't give enough info about the status of the culture. This is particularly important because the lower growth rate in the N-limited culture implies that this experiment was truly RATE limited during 'log' phase by N availability. If this is true, it means the culture should have had a constantly-decreasing growth rate as N drawdown occurred, not a single log-linear rate. Make sure the reader is clear on how these cultures were limited and how growth proceeded - i.e. the difference between N being the limiting nutrient in the Redfield sense (determines maximum culture density as opposed to, say, P or vitamins or something) vs N being the rate limiter (growth is limited by the kinetics of N uptake, instantaneous mu would be independent of light intensity, growth would continually slow if grown in batch). At this light level, it's

hard to envision a batch culture that was rate limited by N at its outset, but still could be grown dense enough to get good alkenone isotope measurements, unless these were truly massive experiments. Give us more information. Do we have final nutrient concentrations or any sense of their evolution over the growth curve?

3) Back to the temperature effect, this 'opposite' effect could be VERY useful for determining the mechanism of the temp. effect, as it would seem to indicate it has to be something more subtle than rate dependence on growth temp., or shifting metabolite into structural vs storage products at different levels of stress, ect. It has to be something that COULD vary strain to strain and has an (apparently?) linear response. However, it does not seem likely from the data that it's related to relative abundance of K37:2 and K37:3. Not only is it not at all clear from Figure 1 that the slopes or intercepts of the isolated 2's and 3's values are significantly different from each other (given their variance at a given temp), but it is unclear from the discussion how the authors are suggesting this 'indirect' effect manifests. If they are invoking a Rayleigh-type mechanism (it sounds like they are) where the negative-offset 3's get progressively heavier as they pull from a progressively heavier pool of remaining 2's, then there shouldn't be any 'switchover' of which compound is heavier or lighter, and K37:3 should get heavier, not lighter, as desaturation becomes more complete at low temperatures. I'd like to see a conceptual model with some rough ballpark numbers explaining the theoretical mass balance between 2s, 3s, and 'waste' hydrogen. If the overall slope of integrated K37s is due to the removal of isotopically heavy hydrogen as a 'loss' term from the desaturation, how would one explain the temp effects observed in saturated lipids (16:0 in Zhang et al) and the difference in the signs of the slopes observed for K37s here and by Wolhowe?

4) Back to the nutrient experiments, it seems like a major point that the exponential- to stationary-phase effect appears to be reverse of what's been observed previously. No discussion is made of this, however.

5) On page 9, there is discussion of how desaturation of 18:0 to 18:1 could counteract

the salinity effect. Are you suggesting that the 18:0 to 18:1 ratio is salinity dependent? Because the depletion from desaturation would occur under all conditions.

6) Lastly, and most importantly, the big "sell" of the paper is the determination that lipids synthesized in the chloroplast don't experience salinity effects, and lipids synthesized (or completed) in the cytosol do. However, there are a couple of problems with the authors' argument that this is the case. First of all, the only lipid that A) clearly does not exhibit a significant slope vs salinity at the same time as B) appearing statistically distinct from the slopes of the OTHER lipids measured in the same organism is phytol from E. huxleyi. I. galbana phytol, while apparently not being significantly correlated with salinity, does not to the eye, at least, appear to exhibit a slope that is statistically distinct from that of, say, brassicasterol. It's lack of slope appears to be driven by a single data point. R. lam phytol, of course, DOES correlate with S. I would like a more consistent demonstration/argument that we can say lipids built in the chloroplast show a distinct response from cytosolic products. Adding to this ambiguity is the discussion of the alkenones. On page 10, the authors state that alkenones are synthesized in the chloroplast. On page 14, they state that alkenones are made in the cytosol. The former statement seems most consistent with previous work - note the work of Eltgroth et al., who show alkenones building up as lipid bodies in the chloroplast. If this is true, it undermines the cytosol-vs-chloroplast-salinity-effect argument. If they are produced in the cytosol, they help the argument, but there's no evidence or citation provided to this effect.

In response to the specific questions for assessment:

Does the paper address relevant scientific questions within the scope of BG? Yes.

Does the paper present novel concepts, ideas, tools, or data? Yes.

Are substantial conclusions reached? Yes.

Are the scientific methods and assumptions valid and clearly outlined? The methods

are good, but there are some concerns about the reasoning, as discussed above.

Are the results sufficient to support the interpretations and conclusions? It is currently slightly unclear whether they are or are not. As stated, no, but better explanation of the authors' reasoning may help here.

Is the description of experiments and calculations sufficiently complete and precise to allow their reproduction by fellow scientists (traceability of results)? Mostly, but as noted above some additional info about the cultures would be appreciated.

Do the authors give proper credit to related work and clearly indicate their own new/original contribution? It is at times difficult to keep track of what data comes from wholly new experiments, what data was collected in the course of other studies, and what data is NEW data but collected in the COURSE of those other studies. Credit is given, yes, but organization' of the various "classes" of data could be better to avoid the appearance of 'double publishing'.

Does the title clearly reflect the contents of the paper? Yes

Does the abstract provide a concise and complete summary? Yes.

Is the overall presentation well structured and clear? The logical flow of the discussion needs work. It should be clear from the outset what the problem is and the facts presented in a clear progression to lead the reader to the conclusion. No parts are wholly inappropriate, but for example the discussion of 1) NADPH sources, then 2) the discussion of H-water exchange, then 3) the discussion of osmolytes and ten 4) the (paraphrased) statement "as you can see NADPH sources are what control things" makes it seem as though 2 and 3 were just inserted after the fact without any consequence to the narrative. Transition sentences between sections/paragraphs would help.

Is the language fluent and precise? Yes. This is refreshingly polished for a review manuscript!

Are mathematical formulae, symbols, abbreviations, and units correctly defined and used? Yes.

Should any parts of the paper (text, formulae, figures, tables) be clarified, reduced, combined, or eliminated?

We really don't need both dD vs_____ and alpha vs _____ plots, at BEST they show the same thing, and if they don't it's because dD water is varied and dD plots are useless.

Are the number and quality of references appropriate? Yes.

Is the amount and quality of supplementary material appropriate? Mostly. I sure would like some growth data.

---

## Referee Comment (RC2) · Anonymous Referee #2 · 4 Jun 2019

This paper combined hydrogen isotope salinity data from alkenones (previously published) with hydrogen isotope salinity data from additional lipids (some previously published), also new temperature and nitrogen culture data for alkenones (C37). The data suggest that increased temperature may cause C37 2H-enrichment, confirms that higher growth rates (achieved through different media N levels) leads to increased C37 fractionation, and confirms that other lipid classes (not just C37) in 2 haptophyte groups also become 2H-enriched at higher salinity (but not phytol in 2 species). I especially appreciate the measurement of several different lipid classes. Alkenones have been the sole objective of many previous studies – but ignoring the other lipid classes restricts the potential for understanding the fractionation mechanisms (in haptophytes and other

species). The isotopic responses of non-alkenone lipids to environmental variations in culture are inherently fascinating in their own right and add valuable insight into the innerworkings of microbes and their isotopes. Please, tell all your friends, measure the other lipids too – it is worth the instrument time. With that said, it would be great if the nutrient and temperature part could include other lipids besides just C37.

Despite the potential of the paper and the quality of the data, the flow of the paper is currently difficult to follow, and the discussion arguments seem like they are not fully thought out. I offer specific comments below that should hopefully help improve the manuscript, but suggest a major re-working of the structure and perhaps framing of the manuscript. I don't see why the authors want to combine the new temp/nutrient experiments with salinity data (maybe they are not enough for a stand-alone manuscript?) but as is, these aspects don't do a good job supporting one another in a comprehensible story. They seem disjointed and unrelated. One suggestion is to tell the reader why these two findings are combined in a single manuscript – how do they support each other, and what new insight can be gained from putting them both here? If it just doesn't work – maybe they should be separate. Finally, since the time this paper was submitted, a new D/H NADPH paper has been published. It might help streamline or motivate the discussion: www.pnas.org/cgi/doi/10.1073/pnas.1818372116

Title - "metabolic pathways" should be replaced with "lipid biosynthesis pathways" or at least "lipid metabolism" because there are so many things associated with metabolism (but not directly related to lipid biosynthesis) that could potentially impact lipid isotope ratios (or not affect them at all). As it stands, your title doesn't capture the added contribution of various lipid classes that this paper has to offer, it would be great if it could. Additionally, why ignore the temp and nutrient data in the title?

Abstract - Line 27: Again, the word metabolism is too vague here. I think "location of lipid synthesis" would be more specific and thus more helpful for readers to follow your meaning. While the abstract successfully and clearly explains the results, it ends abruptly and the opportunity to add the "so what" part to your paper is lost. Are you

excited about knowing a little bit more about the mechanism? Is it important that not all lipids respond equally to salinity. . ..what are the implications to the biogeosciences? I would pick a motivating point and wrap up the abstract with something that will make the reader want to read more.

Introduction - Line 12: Sorry if I am wrong about this, but would be worth checking if C. tobin is in Group 1. DOI:10.1371/journal.pgen.1005469 Page 2 Line 15: Sachs and Kawka 2015 is not an appropriate reference here as they don't experiment with salinity. Since you are including field studies (sachse et al. 2012) you might also mention studies that came out after 2012 (ie http://dx.doi.org/10.1016/j.gca.2014.03.007 ).

Methods - It isn't mentioned anywhere that fatty acids were corrected for added H from methylation or sterol/phytol corrected for acetylation. I am assuming this was done? If it wasn't, please do so and update data/tables/graphs as necessary. Page 3 Line 21: Why are you calling the sterol/phytol fraction the polar fraction? Page 4 Line 6: Were the fatty acids extracted from the other half of the TLE? This is unclear Page 4 Line 9: Please provide the nutrient recipe(s) Page 5 Line 10-11. H2 gas was only used to monitor machine accuracy? H2 gas at beginning and end of sequence needs to be used to tie the Isodat software calculations as well, how else are you getting Isodat to correct?

Results - It isn't clear until the Results section that all of the C37 data built up in the introduction is actually from other studies. Maybe earlier you can clarify what exactly you are adding to previous C37 data. The supplement table really helps to do this, perhaps is should be in the main part of the paper. Page 5 Line 26: since you used artificial seawater, can't you just measure your lab's water and estimate alpha with some reasonably big error bars - if you don't know the month it was collected, analyze samples from each month Page 6 Line 14: by "nutrients" don't you just mean "the effect of nitrogen limitation"? Please use more specific language Page 7 Line 4. I think this is supposed to be section 3.2 (not 3.1)

Discussion - 4.1 – There are 3 issues. Firstly, it was claimed that this temperature part was of secondary interest earlier in the paper, and yet it is the leading discussion point. Either move this down or change the framing of the paper. Secondly, a tremendous amount of text was devoted to invoking abundance shifts in alkenone type to explain the temp trend but no graph (either data or schematic) is offered to support this interpretation. (Along those lines, it is always interesting to show how UK37 does in temperature experiments, even if just supplementary. It would be worth reporting how well this strain does at reconstructing temperature when grown in controlled temperature conditions.) Thirdly, the final sentence is confusing – how is invariable alkenone concentration evidence that growth rate didn't impact 2H/1H ratios? And do you mean total alkenone concentration? B/c most of this section eludes to alkenone abundance changes. Page 7 Line 20-21. How does it compare to the other microbe temp-D/H studies? (Dirghangi and Pagani 2013 http://dx.doi.org/10.1016/j.orggeochem.2013.09.007 & http://dx.doi.org/10.1016/j.gca.2013.05.023 and Zhang et al. 2009 doi:10.1016/j.orggeochem.2008.11.002 ) Page 7 Line 23: Please report somewhere the entire significant positive correlation (with slope, intercept, and their standard errors) for this and other relationships reported in this paper. Maybe just a table or on the graph would be fine if it fits.

4.2 – Line 25 a reference is missing here (Sachs and Kawka 2015) Same comment about section 4.1 apply regarding the framing of the paper. Both sections never really get around to the "so what" part and neither does the conclusion. Please, tell us what is the purpose of these sections – how do they add to the story and why are they important? It would make a little more sense if section 4.1 and 4.2 also included non alkenone data, but as is they really stick out.

4.3 – if you really want this to be the main point of your paper, you should address it first in your discussion Line 11 – "in" not "Impact of salinity "of" haptophyte lipids"

4.3.1 - Page 9 Line 21 – somewhere around here would be a good place to compare the lack of C16:0 EHUX correlation with the strong relationship found in Sachs et al.

2016 Page 10 Line 1 – “values” is misspelled Page 10 – Lines 12-14. This is extremely misleading. Plenty of pyruvate is also made in the chloroplast (as the paper mentions later on). Furthermore, Acetyl-CoA is not known to pass organelle walls according to several plant biochem text books. DeNiro and Epstein is not an appropriate reference for this – instead you should check Lohr et al. 2012 (10.1016/j.plantsci.2011.07.018 ) and Hemmerlin et al. 2012 (10.1016/j.plipres.2011.12.001 ) even though they focus on sterols, it is clear that pyruvate can be made in the cholorplast. Certainly under some conditions algal pyruvate seems to be imported into the chloroplast (DOI:10.1371/journal.pgen.1006490) but it is incorrect to leave your statement as is. Page 11 Line 7. Incorrect information, actually the diatom sterol was highly affected by light intensity, strikingly in the opposite manner as phytol and the C14:0 fatty acid. This mistake, and the interpretation that depends on it needs to be fixed.

4.4 - This section would greatly benefit from some rearrangement and reworking to help the reader. It is difficult to follow. One way to improve this is add a brief outline of the points you want to make in the first paragraph before hitting on all of them. A schematic would also help. Are you suggesting anything new here or just reporting all the previously suggested hypotheses? There is no need to devote so much text to explaining these previous hypotheses, a short summary sentence for each should do. Isn’t there something more unique you can add now that you have this extra data from the other lipids? Isn’t it significant that several studies now have seen only a weak (or no) relationship with phytol? One of the main issues with the NADPH (OPP vs PS1) hypothesis is that NADPH isn’t known to cross organelle walls. Is there an OPP pathway inside the chloroplast in haptophytes? If you want to rely so heavily on this explanation, some evidence (in the form of a citation) for 1) NADPH crossing the membrane or 2) OPP in the chloroplast is really needed here.

FIGURES - While the figures indicate in the caption where previous data is coming from, it would be helpful if this info was more visually accessible in the key, either next to species names if no regression is given (Fig 3 and 4), or, next to regressions that

should be provided (full equations) (Fig 4). Some figures have regression lines some don't. Is there a purpose to this?

Fig. 4 - If a relationship isn't significant (phytol) don't add a regression line. . .or do something like make regression lines for significant regressions solid lines and not signification regressions dotted. C16:0 symbol colors and shape are too similar to phytol's.

---

## Author Comment (AC1) · 7 Jun 2019

Anonymous Referee #1 – The manuscript by Weiss et al. reports new isotopic compositional data for alkenones, fatty acids, a sterol, and phytol for three different alkenone producers grown under Sand nutrient- experimental conditions. The primary novel contributions the data makes are the interesting responses of alkenone dD to both T and growth phase, in opposition to what has been reported in the literature for other alkenone-producers. They also further document the "salinity effect" on hydrogen isotope fractionation in a number of lipids produced under varying experimental conditions. While the new data is well worth reporting, and may tell very interesting stories

about the potential mechanisms behind the temperature, growth rate/phase, and salinity effects, the manuscript as it currently stands has several flaws in its arguments. The data quality, and overall writing, are worthy of publication - the well-documented experimental section is particularly appreciated. However, major revisions to the overall argumentative thread of the paper will be necessary before this can be vetted.

In no particular order, here are some concerns with the paper that need to be addressed:

RESPONSE: We would like to thank anonymous referee 1 for their thoughtful feedback on our manuscript and address their major concerns as "RESPONSE: " following the original comment.

1) We really need the T. lutea growth rates for the temp experiments. It is true that, depending on culture density, you may have divergence of chlorophyll fluo. growth rate from cell count growth rate, due to shading, but it is still better than nothing. Given that the temperature effect is completely the opposite of what has been seen before (note that in addition to the somewhat-indeterminate alkenone work, the negatively sloped temperature effect is also seen in other lipids by Zhang et al. 2009, Organic Geochem), and given the growth rate effects shown here and elsewhere, an attempt should be made at least to constrain how much temp-dependent rate change may impact (counteract?) this curve. The note about the approximately-identical per-volume alkenone concentrations is potentially useful, but only if the cultures were all inoculated at exactly the same density, took off identically with identical lag phases, etc. F curves would be more useful.

2) We also need to see growth curves for the nutrient experiment (f-based or otherwise). "Day 4 and day 10" doesn't give enough info about the status of the culture. This is particularly important because the lower growth rate in the N-limited culture implies that this experiment was truly RATE limited during 'log' phase by N availability. If this is true, it means the culture should have had a constantly-decreasing growth rate
as N drawdown occurred, not a single log-linear rate. Make sure the reader is clear on how these cultures were limited and how growth proceeded - i.e. the difference between N being the limiting nutrient in the Redfield sense (determines maximum culture density as opposed to, say, P or vitamins or something) vs N being the rate limiter (growth is limited by the kinetics of N uptake, instantaneous mu would be independent of light intensity, growth would continually slow if grown in batch). At this light level, it's hard to envision a batch culture that was rate limited by N at its outset, but still could be grown dense enough to get good alkenone isotope measurements, unless these were truly massive experiments. Give us more information. Do we have final nutrient concentrations or any sense of their evolution over the growth curve?

RESPONSE: Referee #1 inquired about growth rate information (points 1 and 2), which we provide now as Figs. 1 and 2 in this comment. For the temperature experiment, we only have the chlorophyll fluorescence data, but for the nutrient experiment, we have cell count data. Growth rates for both experiments follow the traditional pattern of exponential to stationary growth reported in previous haptophyte culture experiments. N was the limiting nutrient here in our nutrient experiments, leading to slower growth for the N-reduced relative to N-replete batches.

3) Back to the temperature effect, this 'opposite' effect could be VERY useful for determining the mechanism of the temp. effect, as it would seem to indicate it has to be something more subtle than rate dependence on growth temp., or shifting metabolite into structural vs storage products at different levels of stress, ect. It has to be something that COULD vary strain to strain and has an (apparently?) linear response. However, it does not seem likely from the data that it's related to relative abundance of K37:2 and K37:3. Not only is it not at all clear from Figure 1 that the slopes or intercepts of the isolated 2's and 3's values are significantly different from each other (given their variance at a given temp), but it is unclear from the discussion how the authors are suggesting this 'indirect' effect manifests. If they are invoking a Rayleigh-type mechanism (it sounds like they are) where the negative-offset 3's get progressively heavier BGD
as they pull from a progressively heavier pool of remaining 2's, then there shouldn't be any 'switchover' of which compound is heavier or lighter, and K37:3 should get heavier, not lighter, as desaturation becomes more complete at low temperatures. I'd like to see a conceptual model with some rough ballpark numbers explaining the theoretical mass balance between 2s, 3s, and 'waste' hydrogen. If the overall slope of integrated K37s is due to the removal of isotopically heavy hydrogen as a 'loss' term from the desaturation, how would one explain the temp effects observed in saturated lipids (16:0 in Zhang et al) and the difference in the signs of the slopes observed for K37s here and by Wolhowe?

RESPONSE: We agree that a Rayleigh fractionation mechanism might not be the best way to explain the data from the temperature experiment. van der Meer et al. (2013) explained the offset between the two alkenones (Dd2H) in this manner. Both the Dd2H vs UK'37 (Fig. 3a) and the UK'37 vs temperature (Fig. 3b) suggest that temperature likely does have some effect on 2H ratios of alkenones. When we plot our data on top of the van der Meer et al. (2013) compilation of Dd2H vs UK'37, we observe much more scatter / outliers, especially from our temperature experiment, suggesting that temperature alone cannot explain this. The effects of desaturation might be dampening / overwriting the temperature correlation with 2H ratios, especially at the extreme high and low ends of the temperature range. The situation is further complicated by the fact that the C37:3 is synthesized from a pool of C37:2 which is also still being synthesized, so not a fixed source. The influence of temperature on 2H ratios of alkenones is still unclear, and as mentioned, other compounds show the opposite of what we report here. However, we highlight here that temperature likely exhibits an effect on hydrogen isotope ratios of alkenones and this effect does not appear to be uniform across experiments. Ultimately, this issue would greatly benefit from testing in a chemostat culture to remove any effects associated with growth phase and growth rate. Analysis of a suite of compounds from such a chemostat could also help to understand these biosynthetic differences in further detail.

**BGD**
4) Back to the nutrient experiments, it seems like a major point that the exponential- to stationary-phase effect appears to be reverse of what's been observed previously. No discussion is made of this, however.

RESPONSE: In our nutrient experiment, there was greater accumulation of alkenones during stationary phase under both nutrient concentrations, and a previous study showed a greater concentration of alkenones per cell in stationary and decline phases for other Group II species I. galbana and R. lamellosa (Chivall et al., 2014, GCA). These previous growth phase experiments measuring 2H ratios of alkenones (Chivall et al., 2014) showed a decrease in sensitivity to salinity during stationary and death phases, relative to exponential growth. There was depletion in 2H (lower alpha, more fractionation) associated with longer growth in I. galbana, but no difference was seen between phases for R. lamellosa (potentially as a result of cell clumping). We also noted a depletion in 2H with longer growth, similar to I. galbana. Thus, while accumulation of alkenones might be different, the fractionation response to growth phase appears to be similar.

5) On page 9, there is discussion of how desaturation of 18:0 to 18:1 could counteract the salinity effect. Are you suggesting that the 18:0 to 18:1 ratio is salinity dependent? Because the depletion from desaturation would occur under all conditions.

RESPONSE: We are not suggesting that desaturation from 18:0 to 18:1 is salinity dependent. Instead we hypothesize that the hydrogen isotope fractionation related to this desaturation step might be large enough to mask the salinity effect. We will clarify this in a revised version.

6) Lastly, and most importantly, the big "sell" of the paper is the determination that lipids synthesized in the chloroplast don't experience salinity effects, and lipids synthesized (or completed) in the cytosol do. However, there are a couple of problems with the authors' argument that this is the case. First of all, the only lipid that A) clearly does not exhibit a significant slope vs salinity at the same time as B) appearing statistically
distinct from the slopes of the OTHER lipids measured in the same organism is phytol from E. huxleyi. I. galbana phytol, while apparently not being significantly correlated with salinity, does not to the eye, at least, appear to exhibit a slope that is statistically distinct from that of, say, brassicasterol. It's lack of slope appears to be driven by a single data point. R. lam phytol, of course, DOES correlate with S. I would like a more consistent demonstration/argument that we can say lipids built in the chloroplast show a distinct response from cytosolic products. Adding to this ambiguity is the discussion of the alkenones. On page 10, the authors state that alkenones are synthesized in the chloroplast. On page 14, they state that alkenones are made in the cytosol. The former statement seems most consistent with previous work - note the work of Eltgroth et al., who show alkenones building up as lipid bodies in the chloroplast. If this is true, it undermines the cytosol-vs-chloroplast-salinity-effect argument. If they are produced in the cytosol, they help the argument, but there's no evidence or citation provided to this effect.

RESPONSE: Eltgroth et al. (2005) suggest that PULCA are associated with the chloroplast, but also the endoplasmic reticulum. Sawada and Shiraiwa (2004) report alkenones are found in the ER and the coccolith producing vesicle. Eltgroth et al. (2005) suggest that their chloroplast fraction likely included ER components. Haptophytes have a peripheral ER (Andersen, 2004). A peripheral ER has a connection with the cytosol (English et al., 2009), thus our main argument remains true. If alkenones are indeed present / stored in the ER, (as well as other organelles), they could be associated with the chloroplast, but they are also connected to the cytosol. Furthermore, both Eltgroth et al.(2005) and Sawada and Shiraiwa (2004) show where alkenones are accumulating, not where they are produced. It is possible that alkenones are produced elsewhere and then stored in lipid bodies in various locations in the cell. Additionally, if alkenones are synthesized from fatty acids, this might occur immediately in the ER where fatty acid elongation and desaturation takes place (Jónasdóttir, 2019), or potentially at a later stage from excess fatty acids which have been exported to the cytosol. With respect to phytol, Sachs et al. (2016) do not show a significant change in 2H with

**BGD**
salinity, similar to what we observe here. We strive to clarify our argument in a revised manuscript by emphasizing the E.R. in the discussion, which has a connection to the cytosol compared to the chloroplast, which is closed.

BGD
Fig. 1. Growth curves for the Temperature experiment based on chlorophyll fluorescence.

---

## Referee Comment (RC3) · Anonymous Referee #3 · 11 Jun 2019

The manuscript by Weiss et al. set up quite ambitious goals to address almost all factors affecting D/H fractionation in haptophyte lipids. For that purpose the authors included quite a bit previous published data. However, they were not mentioned until Section 3, Results. Through the Introduction and Method, as well as in figures, one can hardly tell what are new and what were previously published. In addition, the new and old dataset seem disjointed each other and it seemed to me that such a way of merging data helped to demonstrate a comprehensible story. Though the data presented are interesting and potentially valuable, the manuscript as written suffered many fatal deficiencies. It is very difficult to follow the flow of the manuscript. The arguments in the Discussion section were not well organized and demonstrated. I

would suggest resubmission after a complete overhaul.

It is a big headache to follow the Method section. I had to list the details of all different cultures on a piece of paper to sort out all different parameters. They were in such a mess: different media, seawater (artificial seawater and filtered seawater), growth temperature (E. huxleyi, E. glabana and R. lamellosa group set up at 15°C, T. lutea strain CCAP 927/14 culture at 20-23°C, and T. lutea strain CCAP 463 and NIES-2590 cultures at 10-35°C), light intensity (60, 100, and 180-220 $\mu$mol photons m-2 s-1), growth phases for collection (linear, exponentially, and stationary), measurements of growth rate (chlorophyll fluorescence, and daily cell counts), and even GC columns (leading to integrated C37 peaks or individual C37:2/C37:3 peaks)....It would be hard to imagine if anyone else could come up with a more complicated and confusing experiment design than this one. Such awful setup simply made it hard to isolate one single variable and the arguments based on such data less convincing.

I was curious why the authors did not give any description of methylation of fatty acids or acetylation of sterols, as they are essential to figure out how reliable their reported dD data of fatty acids and sterols. Neither did they present a GC-IRMS trace to demonstrate how well peaks were separated, as sterols often co-eluted. Those are essential to evaluate the data quality.

Another fundamental flaw was that not even a single growth curve was presented, given the fact harvest was taken in different growth phases, and temperature would impact the growth rate. In particular, the authors wanted to address the effect of nutrient replete (NR) and nutrient limited (NL) on lipid D/H fractionation. For this purpose, it would be essential to know how growth rate changed daily. When the authors stated NR or NL, only nitrate concentrations were given, but no phosphorous concentrations— this set of experiments were performed using filtered seawater but no information was available regarding N/P ratio in NR and NL cultures. Table 2 did not give the date for the presented division rate, Day 4 or D10—in fact the selection of the date seemed randomly. Indeed no one knows what happened between Day 1 to Day 4, or between

Day 4 and Day 10, as far as the status of culture is concerned. As a result we actually don't know when nutrient availability BEGAN to limit the growth rate! Presumably at the given light intensity and initial nitrate concentrations, there shall be no limit on growth rate solely by nitrogen availability at the onset. Then it would be essential to know when the rates in NR and NL cultures began to differ and what could cause the difference. These were batch cultures, not chemostatic cultures (Zhang et al., 2009; Organic Geochemistry). As a result, demonstration of growth rate during the log phase truly limited by nitrate availability would be the key. Without growth curves, one was not in a position to address the effect of growth rate. In fact, it would ideal for the authors to give the concentration of individual biomarkers per cell as we can tell if there are any strategic allocation of carbon source or energy during the different growth phases

The title set up two goals to address: 1) impact of metabolic pathways, and 2) salinity on the on the hydrogen isotope ratios of haptophyte lipids. However, the Discussion initiated with temperature effect, followed by nutrient effect. The title seemed misleading. I would hesitate to call them nutrient replete (NR) and nutrient-limited (NL) conditions as they merely differed in nitrate concentration by 0.6 mM, and growth rate by barely half. In Zhang et al. (2009, Organic Geochemistry, doi:10.1016/j.orggeochem.2008.11.002), NR and NL chemostatic cultures differed in nitrate concentrations by almost 70 times and cell division rates by 4.5 times. Even among such huge growth rate differences, fatty acids biosynthesized by acetogenic pathway did not show the difference in D/H fractionation.

In fact the data supporting for argument of temperature and growth rate effects seemed farfetched. Judged by Table 2, I would say the difference was rather small. If we choose dD of C37:3 as an example, the first line for growth rate at 0.14 could give -121-3= -124 ‰ the last line growth rate at 0.21 gave -130+6= -124‰ almost the same. At least such differences were rather small. The same could be found in Table 1 regarding the temperature effect. At least for the batch culture of T. lutea NIES 2590 with salinity varying from 15 to 30, all dD values of C37:3 could be rounded to -140‰ within stan-

Interactive
comment

dard deviation—they were simply the same. It seemed to me that those data would hardly support the argument for the significant positive correlation with temperature, though the phenomena observed differed from reported in Zhang et al. (2009, OG). Maybe D/H fractionation in alkenones much less sensitive to temperature than fatty acids?

The data on salinity effect seemed more robust, at least for R. lamellose and I. galbana (Fig. 1), but it is questionable to say "The $\delta$2HC37 ratios from T. lutea (temperature and nutrient experiments) fit well with values noted for other Group II species I. galbana and R. lamellosa (Fig.3)." (Page 9, Lines 6-7). Such data varied a wide range at a given salinity which could be considered a substantially large standard deviation. Again, there are too many variables influencing D/H fractionation. As a result, such data should not be plotted in Fig. 3. On the other hand the authors should provide full linear equations for R. lamellose and I. galbana under different salinities as the relationship between slope and intercept could help reveal more information.

Section 4.4– Discussion This section needs an overhaul as it is very hard to follow the argument. I would strongly suggest add a schematic figure to demonstrate how biosynthetic pathways affect biomarker D/H fractionation. However, I don't think there were new discoveries here which deserves more than two full pages to elaborate already well known hypothesis. It is well known from previous algal culture experiments that different classes of biomarkers were characterized by substantially different D/H fractionation, in particular, among acetogetic, MVA and DOXP/MEP pathways. On the other hand, the current dataset could not provide sufficient evidence about the OPP pathway supplies a larger portion of NADPH for biosynthesis, as light intensities in the experiments were not low enough.

There are quite a few different families (species) of halophytes. Just wonder if alkenones might be biosynthesized in different organelle among different species. The authors cited (Rontani et al., 2006) to suggest that alkenones are synthesized from these shorter chain fatty acids by elongation and subsequent decarboxylation in the

chloroplast (Page 10, Lines 10-12), but then claimed "Chain elongation leading to long-chain alkenones does take place in the cytosol" (Page 14, Line 4). Previous studies did show different D/H fractionation in biomarkers biosynthesized among different organelle. Would it be possible for difference in alkenone D/H fraction among different families due to different organelle for synthesis of alkenones?

Technical corretions 1. Replace "metabolic" with "biosynthetic" as the paper only address about the biosynthesis of biomarkers. 2. Page 5, lines 8 and 12: "n-alkanes", here "n" should be italic 3. Page 6, Line 22—-There was no Fig. 2c

---

## Author Comment (AC2) · 12 Jun 2019

We would like to thank anonymous referee 2 for their feedback on our manuscript. We propose to fix the grammatical errors in a revised version of the manuscript, and address the more major comments following the word 'RESPONSE' below the original comment.

This paper combined hydrogen isotope salinity data from alkenones (previously published) with hydrogen isotope salinity data from additional lipids (some previously pub-

[Figure]

lished), also new temperature and nitrogen culture data for alkenones (C37). The data suggest that increased temperature may cause C37 2H-enrichment, confirms that higher growth rates (achieved through different media N levels) leads to increased C37 fractionation, and confirms that other lipid classes (not just C37) in 2 haptophyte groups also become 2H-enriched at higher salinity (but not phytol in 2 species). I especially appreciate the measurement of several different lipid classes. Alkenones have been the sole objective of many previous studies – but ignoring the other lipid classes restricts the potential for understanding the fractionation mechanisms (in haptophytes and other species). The isotopic responses of non-alkenone lipids to environmental variations in culture are inherently fascinating in their own right and add valuable insight into the innerworkings of microbes and their isotopes. Please, tell all your friends, measure the other lipids too – it is worth the instrument time. With that said, it would be great if the nutrient and temperature part could include other lipids besides just C37.

Despite the potential of the paper and the quality of the data, the flow of the paper is currently difficult to follow, and the discussion arguments seem like they are not fully thought out. I offer specific comments below that should hopefully help improve the manuscript, but suggest a major re-working of the structure and perhaps framing of the manuscript. I don't see why the authors want to combine the new temp/nutrient experiments with salinity data (maybe they are not enough for a stand-alone manuscript?) but as is, these aspects don't do a good job supporting one another in a comprehensible story. They seem disjointed and unrelated. One suggestion is to tell the reader why these two findings are combined in a single manuscript – how do they support each other, and what new insight can be gained from putting them both here? If it just doesn't work – maybe they should be separate. Finally, since the time this paper was submitted, a new D/H NADPH paper has been published. It might help streamline or motivate the discussion: www.pnas.org/cgi/doi/10.1073/pnas.1818372116

RESPONSE: Indeed, the new paper from Wijker et al. (2019) would be useful to add. They showed that different pathways greatly controlled isotope ratios of NADPH and

therefore lipid 2H ratios, which is similar to what we show here. Genetically, T. lutea is a Group II species, and we wanted to determine whether the 2H ratios were also characteristic of Group II or perhaps showed a more marine 2H signal. We agree that other lipids would add to the interpretation, but at this moment, we do not have that data. The offset between species is important and can potentially help elucidate the salinity mechanism because different pathways might be preferred under salt stress. We will make this clearer in a revised version.

Title - "metabolic pathways" should be replaced with "lipid biosynthesis pathways" or at least "lipid metabolism" because there are so many things associated with metabolism (but not directly related to lipid biosynthesis) that could potentially impact lipid isotope ratios (or not affect them at all). As it stands, your title doesn't capture the added contribution of various lipid classes that this paper has to offer, it would be great if it could. Additionally, why ignore the temp and nutrient data in the title?

RESPONSE: Yes, that is a good point. We propose a new title: Impact of lipid biosynthesis pathways and growth parameters on hydrogen isotope ratios of haptophyte lipids.

Abstract - Line 27: Again, the word metabolism is too vague here. I think "location of lipid synthesis" would be more specific and thus more helpful for readers to follow your meaning. While the abstract successfully and clearly explains the results, it ends abruptly and the opportunity to add the "so what" part to your paper is lost. Are you excited about knowing a little bit more about the mechanism? Is it important that not all lipids respond equally to salinity. . ..what are the implications to the biogeosciences? I would pick a motivating point and wrap up the abstract with something that will make the reader want to read more.

RESPONSE: The important role of specific biosynthetic pathways as a determiner of species-related differences in 2H fractionation is exciting. Understanding these metabolic fluxes and sources of NADPH for specific lipids is therefore very important.

Recent work pointed out above (Wijker et al., 2019) has highlighted the importance of metabolic pathways, specifically for NADPH, on d2H, and our study also suggests that biosynthetic differences in lipid synthesis are important for understanding 2H ratios. We will add a final sentence to the abstract that states: "These findings suggest that not all lipids retain a correlation with salinity, and this appears to be governed by differences in biosynthetic pathways and cellular compartments. Use of lipids relying on a cytosolic connection are more appropriate for applications of 2H ratios to reconstruct salinity in the geologic record."

Introduction - Line 12: Sorry if I am wrong about this, but would be worth checking if C. tobin is in Group 1. DOI:10.1371/journal.pgen.1005469

RESPONSE: To our knowledge, C. tobin does not synthesize alkenones.

Page 2 Line 15: Sachs and Kawka 2015 is not an appropriate reference here as they don't experiment with salinity. Since you are including field studies (sachse et al. 2012) you might also mention studies that came out after 2012 (ie http://dx.doi.org/10.1016/j.gca.2014.03.007 ).

RESPONSE: Yes, you are correct. We will remove the Sachs and Kawka, 2015 reference and add newer references.

Methods - It isn't mentioned anywhere that fatty acids were corrected for added H from methylation or sterol/phytol corrected for acetylation. I am assuming this was done? If it wasn't, please do so and update data/tables/graphs as necessary.

RESPONSE: Yes, fatty acids, sterol and phytol were all corrected for methylation / acetylation. We will clarify this in the methods section.

Page 3 Line 21: Why are you calling the sterol/phytol fraction the polar fraction?

RESPONSE: We separated the TLE into three fractions which we refer to as apolar, ketone and polar fractions.

Page 4 Line 6: Were the fatty acids extracted from the other half of the TLE? This is unclear

RESPONSE: Yes, our traditional separation method using aluminum oxide removes fatty acids, so we separated the TLE into two aliquots before this step. One was used for fatty acids and the other for the alkenones, sterol and phytol.

Page 4 Line 9: Please provide the nutrient recipe(s) Page 5 Line 10-11. H2 gas was only used to monitor machine accuracy? H2 gas at beginning and end of sequence needs to be used to tie the Isodat software calculations as well, how else are you getting Isodat to correct?

RESPONSE: H2 gas was used for Isodat calculations. We will clarify this.

Results - It isn't clear until the Results section that all of the C37 data built up in the introduction is actually from other studies. Maybe earlier you can clarify what exactly you are adding to previous C37 data. The supplement table really helps to do this, perhaps is should be in the main part of the paper.

RESPONSE: We will make it more apparent in the introduction that the alkenone and I. galbana fatty acid data is already published, and move the supplementary table into the main manuscript.

Page 5 Line 26: since you used artificial seawater, can't you just measure your lab's water and estimate alpha with some reasonably big error bars - if you don't know the month it was collected, analyze samples from each month

RESPONSE: since we used extracts from an experiment conducted a few years ago in a different lab, measuring the lab water to calculate alpha for this experiment would be difficult

Page 6 Line 14: by "nutrients" don't you just mean "the effect of nitrogen limitation"? Please use more specific language

RESPONSE: Yes, we will change this.

Page 7 Line 4. I think this is supposed to be section 3.2 (not 3.1)

RESPONSE: Yes, thank you for catching this mistake.

Discussion - 4.1 – There are 3 issues. Firstly, it was claimed that this temperature part was of secondary interest earlier in the paper, and yet it is the leading discussion point. Either move this down or change the framing of the paper. Secondly, a tremendous amount of text was devoted to invoking abundance shifts in alkenone type to explain the temp trend but no graph (either data or schematic) is offered to support this interpretation. (Along those lines, it is always interesting to show how UK37 does in temperature experiments, even if just supplementary. It would be worth reporting how well this strain does at reconstructing temperature when grown in controlled temperature conditions.) Thirdly, the final sentence is confusing – how is invariable alkenone concentration evidence that growth rate didn't impact 2H/1H ratios? And do you mean total alkenone concentration? B/c most of this section eludes to alkenone abundance changes. Page 7 Line 20-21. How does it compare to the other microbe temp-D/H studies? (Dirghangi and Pagani 2013 http://dx.doi.org/10.1016/j.orggeochem.2013.09.007 & http://dx.doi.org/10.1016/j.gca.2013.05.023 and Zhang et al. 2009 doi:10.1016/j.orggeochem.2008.11.002 )

RESPONSE: We propose to restructure the discussion section, and will move the temperature part to later in the discussion. We focused on the temperature effect noted for haptophytes, and therefore did not include other lipids. We will add a supplementary figure showing the correlation between UK'37 and temperature for T. lutea. The final sentence should be revised to: "Total alkenone concentrations did not vary substantially over the temperature range, but faster growth was noted (measured as chlorophyll fluorescence) at higher temperatures. It is possible that growth rate had an effect on $\delta$2HC37 ratios, but since we do not have daily cell counts, we cannot easily compare this effect with previously noted growth rate effects. In previous studies, faster growth

resulted in isotopic depletion, but we note isotopic enrichment for the higher temperatures and faster growing cultures. This isotopic enrichment could be governed by different isoenzymes, which can be associated with different fractionation, operating at higher or lower temperatures, as suggested by Zhang et al. (2009) and Jahnke et al. (1999)."

Page 7 Line 23: Please report somewhere the entire significant positive correlation (with slope, intercept, and their standard errors) for this and other relationships reported in this paper. Maybe just a table or on the graph would be fine if it fits.

RESPONSE: We were discussing the correlation between the alkenone 2H and temperature, not the linear regression equation. We can add the linear regression equation to the graph.

4.2 – Line 25 a reference is missing here (Sachs and Kawka 2015) Same comment about section 4.1 apply regarding the framing of the paper. Both sections never really get around to the "so what" part and neither does the conclusion. Please, tell us what is the purpose of these sections – how do they add to the story and why are they important? It would make a little more sense if section 4.1 and 4.2 also included non alkenone data, but as is they really stick out. 4.3 – if you really want this to be the main point of your paper, you should address it first in your discussion

RESPONSE: We agree that the discussion of the temperature and nutrient experiments should be moved to the end of the discussion. Since we discuss how the effect of 2H fractionation is different lipids for different haptophyte species, we felt it was valuable to include the alkenone data from T. lutea here since there has been no previous characterization of d2H ratios for this alkenone-producing strain. We agree that measurement of other lipids from this strain would be interesting and useful, but at this moment, we do not have that data. Understanding the effect of different biosynthetic pathways on the 2H-salinity relationship is not only interesting, but is important for paleo applications because one major issue for the alkenone paleosalinity proxy is our

lack of understanding of the salinity effect on 2H ratios. Additionally, the fact that this salinity signal is present for other lipids is intriguing and may potentially shed light on how this relationship with salinity actually works. Of course, a more in-depth look into these biosynthetic mechanisms would be better, but some insight can still be gained from our approach here. In sediment samples from the geologic record, an integrated signal incorporating effects from a number of variables in addition to salinity (growth related effects as a result of temperature and nutrient concentrations, light intensity, etc.) and it is important to constrain the impact of these variables on d2H ratios as well, which is why we decided to include T. lutea data in this manuscript. We will add a better comparison of our temperature dataset with previously published data.

Line 11 – "in" not "Impact of salinity "of" haptophyte lipids" 4.3.1 - Page 9 Line 21 – somewhere around here would be a good place to compare the lack of C16:0 EHUX correlation with the strong relationship found in Sachs et al. 2016

RESPONSE: We will add this.

Page 10 Line 1 – "values" is misspelled Page 10 – Lines 12-14. This is extremely misleading. Plenty of pyruvate is also made in the chloroplast (as the paper mentions later on). Furthermore, Acetyl-CoA is not known to pass organelle walls according to several plant biochem text books. DeNiro and Epstein is not an appropriate reference for this – instead you should check Lohr et al. 2012 (10.1016/j.plantsci.2011.07.018 ) and Hemmerlin et al. 2012 (10.1016/j.plipres.2011.12.001 ) even though they focus on sterols, it is clear that pyruvate can be made in the cholorplast. Certainly under some conditions algal pyruvate seems to be imported into the chloroplast (DOI:10.1371/journal.pgen.1006490) but it is incorrect to leave your statement as is.

RESPONSE: Thank you for these references. Pyruvate can be imported and exported from the chloroplast, and therefore used in both compartments for formation of acetyl-CoA. We are not suggesting that acetyl-CoA is passing organelle walls. Our thinking here was that if pyruvate originally formed in the cytosol entered the chloroplast and

was used in fatty acid synthesis, this might explain why some fatty acids show a correlation to salinity despite being (mostly) synthesized in the chloroplast. We will rephrase this sentence.

Page 11 Line 7. Incorrect information, actually the diatom sterol was highly affected by light intensity, strikingly in the opposite manner as phytol and the C14:0 fatty acid. This mistake, and the interpretation that depends on it needs to be fixed.

RESPONSE: Yes, thank you for catching this. We will revise this discussion to address this.

4.4 - This section would greatly benefit from some rearrangement and reworking to help the reader. It is difficult to follow. One way to improve this is add a brief outline of the points you want to make in the first paragraph before hitting on all of them. A schematic would also help. Are you suggesting anything new here or just reporting all the previously suggested hypotheses? There is no need to devote so much text to explaining these previous hypotheses, a short summary sentence for each should do. Isn't there something more unique you can add now that you have this extra data from the other lipids? Isn't it significant that several studies now have seen only a weak (or no) relationship with phytol? One of the main issues with the NADPH (OPP vs PS1) hypothesis is that NADPH isn't known to cross organelle walls. Is there an OPP pathway inside the chloroplast in haptophytes? If you want to rely so heavily on this explanation, some evidence (in the form of a citation) for 1) NADPH crossing the membrane or 2) OPP in the chloroplast is really needed here.

RESPONSE: Cormier et al. (2018) showed the OPP in both the chloroplast and cytosol (Fig. 4), and Sachs et al. (2016) discuss presence of OPP in the cytosol. It is well known for diatoms that OPP is located in the cytosol and the same is thought to be the case for haptophytes. We think our hypothesis is still sound with respect to OPP vs PS1. We think it is significant that phytol generally has a weak correlation with salinity. This is what led us to the separation of lipid synthesis compartments with respect

to the salinity correlation. Phytol is different than these other lipid biomarkers being assessed because it is entirely synthesized in the chloroplast and is a component of chlorophyll, and strongly correlated to light intensity (Sachs et al., 2017). We agree that numbering the hypotheses for salinity mechanisms would be beneficial and help make the discussion more succinct. We see why you might be confused about the OPP vs PS1 hypothesis, and we are not proposing the NADPH would cross organelle walls. What we propose is that under certain conditions, OPP or PS1 might be more or less active. If isoprenoid precursors are synthesized by either MVA or MEP, these pathways are located in separate compartments and would derive NADPH from either OPP or PS1 respectively, and thus lipids would be influenced by this difference in activity of OPP or PS1.

FIGURES - While the figures indicate in the caption where previous data is coming from, it would be helpful if this info was more visually accessible in the key, either next to species names if no regression is given (Fig 3 and 4), or, next to regressions that should be provided (full equations) (Fig 4). Some figures have regression lines some don't. Is there a purpose to this? Fig. 4 - If a relationship isn't significant (phytol) don't add a regression line. . .or do something like make regression lines for significant regressions solid lines and not signification regressions dotted. C16:0 symbol colors and shape are too similar to phytol's.

RESPONSE: We did not include regression lines for Figure 2 because of the low number of data points. We will add regression lines to Figure 3 and take your suggestion to make different regression lines when the relationship is significant. We will add publication data into the figure.

---

## Author Comment (AC3) · 2 Jul 2019

We would like to thank anonymous referee 3 for their feedback on our manuscript. We address their comments following the word 'RESPONSE' below the original comment.

The manuscript by Weiss et al. set up quite ambitious goals to address almost all factors affecting D/H fractionation in haptophyte lipids. For that purpose the authors included quite a bit previous published data. However, they were not mentioned until Section 3, Results. Through the Introduction and Method, as well as in figures, one

can hardly tell what are new and what were previously published. In addition, the new and old dataset seem disjointed each other and it seemed to me that such a way of merging data helped to demonstrate a comprehensible story. Though the data presented are interesting and potentially valuable, the manuscript as written suffered many fatal deficiencies. It is very difficult to follow the flow of the manuscript. The arguments in the Discussion section were not well organized and demonstrated. I would suggest resubmission after a complete overhaul.

RESPONSE: We will specify the different experiments in the introduction, and make sure to clarify when we discuss our new results and already published results.

It is a big headache to follow the Method section. I had to list the details of all different cultures on a piece of paper to sort out all different parameters. They were in such a mess: different media, seawater (artificial seawater and filtered seawater), growth temperature (E. huxleyi, E. glabana and R. lamellosa group set up at  $15\hat{a}U\bar{e}C$ , T. lutea strain CCAP 927/14 culture at 20- $23\hat{a}U\bar{e}C$ , and T. lutea strain CCAP 463 and NIES-2590 cultures at 10- $35\hat{a}U\bar{e}C$ ), light intensity (60, 100, and 180-220  $\mu$ mol photons m-2 s-1), growth phases for collection (linear, exponentially, and stationary), measurements of growth rate (chlorophyll fluorescence, and daily cell counts), and even GC columns (leading to integrated C37 peaks or individual C37:2/C37:3 peaks). . ...It would be hard to imagine if anyone else could come up with a more complicated and confusing experiment design than this one. Such awful setup simply made it hard to isolate one single variable and the arguments based on such data less convincing.

RESPONSE: The reason it seems complicated is because these were multiple, separate batch culture experiments conducted at different times. It was not one big experiment. There were five experiments in total. Three separate experiments of E. huxleyi, R. lamellosa and I. galbana respectively grown over a range of salinities, one batch culture of T. lutea grown over a range of temperatures, and one T. lutea batch at two different nutrient concentrations. We will revise the methods section to make it clear that the experiments were separate.

**BGD**
I was curious why the authors did not give any description of methylation of fatty acids or acetylation of sterols, as they are essential to figure out how reliable their reported dD data of fatty acids and sterols. Neither did they present a GC-IRMS trace to demonstrate how well peaks were separated, as sterols often co-eluted. Those are essential to evaluate the data quality.

RESPONSE: Fatty acids, sterol and phytol were all corrected for methylation / acetylation. We will add this information to section 2.2: Fatty acids were derivatized by methylation as described by Heinzelmann et al. (2015), and corrected for the addition of methyl hydrogen (d2HME =  $-171 \pm 1$  ‰. The fraction containing brassicasterol and phytol was acetylated using acetic anhydride following Das and Chakraborty (2011), and d2H of both lipids were corrected for addition of hydrogens (d2H of acetic anhydride = -126 ‰.

Another fundamental flaw was that not even a single growth curve was presented, given the fact harvest was taken in different growth phases, and temperature would impact the growth rate. In particular, the authors wanted to address the effect of nutrient replete (NR) and nutrient limited (NL) on lipid D/H fractionation. For this purpose, it would be essential to know how growth rate changed daily. When the authors stated NR or NL, only nitrate concentrations were given, but no phosphorous concentrations— this set of experiments were performed using filtered seawater but no information was available regarding N/P ratio in NR and NL cultures. Table 2 did not give the date for the presented division rate, Day 4 or D10 in fact the selection of the date seemed randomly. Indeed no one knows what happened between Day 1 to Day 4, or between Day 4 and Day 10, as far as the status of culture is concerned. As a result we actually don't know when nutrient availability BEGAN to limit the growth rate! Presumably at the given light intensity and initial nitrate concentrations, there shall be no limit on growth rate solely by nitrogen availability at the onset. Then it would be essential to know when the rates in NR and NL cultures began to differ and what could cause the difference. These were batch cultures, not chemostatic cultures (Zhang et al., 2009; Organic GeochemInteractive comment

istry). As a result, demonstration of growth rate during the log phase truly limited by nitrate availability would be the key. Without growth curves, one was not in a position to address the effect of growth rate. In fact, it would ideal for the authors to give the concentration of individual biomarkers per cell as we can tell if there are any strategic allocation of carbon source or energy during the different growth phases

RESPONSE: We will include growth curves in a revised version. See Figures 1 and 2 below. It is true, growth was not limited by N availability at the onset. Growth rates for the two nutrient experiments started to differ on the second day of growth, as you can see in the figure 2 below. We will provide concentration data for the alkenones from T. lutea, but not for the other lipids.

The title set up two goals to address: 1) impact of metabolic pathways, and 2) salinity on the on the hydrogen isotope ratios of haptophyte lipids. However, the Discussion initiated with temperature effect, followed by nutrient effect. The title seemed misleading. I would hesitate to call them nutrient replete (NR) and nutrient-limited (NL) conditions as they merely differed in nitrate concentration by 0.6 mM, and growth rate by barely half. In Zhang et al. (2009, Organic Geochemistry, doi:10.1016/j.orggeochem.2008.11.002), NR and NL chemostatic cultures differed in nitrate concentrations by almost 70 times and cell division rates by 4.5 times. Even among such huge growth rate differences, fatty acids biosynthesized by acetogenic pathway did not show the difference in D/H fractionation.

RESPONSE: We will reorganize the discussion to start with the salinity effect and discuss the T. lutea data at the end. For the nutrient experiment, the offsets between exponential and stationary phase are not large, around 6 ‰ which is near machine precision. However, the measured difference in d2Hlipd between the cultures with different nutrient concentrations is larger, 12-13 ‰ and this offset is present for both phases of growth. As you mention, the differences in nutrient concentration are not large, but still we see a depletion at higher nutrient concentration for both growth phases, and this depletion is larger than machine error, so we think this is a real signal. While Zhang et

**BGD**
al. (2009) did not observe a difference for fatty acids at the different nutrient concentrations, there was a difference observed for sterols, an enrichment of approximately 30 ‰ at lower nitrate concentrations, and we note the same trend for long-chain alkenones. It might be that the effects of nutrients and growth rate are also related to biosynthesis and cellular compartment.

In fact the data supporting for argument of temperature and growth rate effects seemed farfetched. Judged by Table 2, I would say the difference was rather small. If we choose dD of C37:3 as an example, the first line for growth rate at 0.14 could give -121-3= - 124 ‰ the last line growth rate at 0.21 gave -130+6= -124‰ almost the same. At least such differences were rather small. The same could be found in Table 1 regarding the temperature effect. At least for the batch culture of T. lutea NIES 2590 with salinity varying from 15 to 30, all dD values of C37:3 could be rounded to -140‰ within standard deviation they were simply the same. It seemed to me that those data would ËĞ hardly support the argument for the significant positive correlation with temperature, though the phenomena observed differed from reported in Zhang et al. (2009, OG). Maybe D/H fractionation in alkenones much less sensitive to temperature than fatty acids?

RESPONSE: For the temperature experiment, it is true that the differences are not large between the different temperatures. However, the overall trend is enrichment, which is different from previous results. There is a strong, positive correlation between d2H and temperature (r = 0.80, p

salinity which could be considered a substantially large standard deviation. Again, there are too many variables influencing D/H fractionation. As a result, such data should not be plotted in Fig. 3. On the other hand the authors should provide full linear equations for R. lamellose and I. galbana under different salinities as the relationship between slope and intercept could help reveal more information.

RESPONSE: The variation seen for the T. lutea data at a salinity of 30 (Figure 3a in the manuscript) is caused by the temperature experiment, so it is actually the variation in temperature that you see there. There are a lot of factors influencing fractionation, however, species and salinity seem to be the two most robust effects. There is approximately a 100 ‰ difference between alkenones from Group II and Group III species, with Group II species being more enriched. The fact that T. lutea data falls in the range of other Group II species is important because it adds further evidence to support the observation that Group II and III species fractionate differently. We shall add the linear regression information for R. lamellosa and I. galbana to Figure 3.

Section 4.4– Discussion This section needs an overhaul as it is very hard to follow the argument. I would strongly suggest add a schematic figure to demonstrate how biosynthetic pathways affect biomarker D/H fractionation. However, I don't think there were new discoveries here which deserves more than two full pages to elaborate already well known hypothesis. It is well known from previous algal culture experiments that different classes of biomarkers were characterized by substantially different D/H fractionation, in particular, among acetogetic, MVA and DOXP/MEP pathways. On the other hand, the current dataset could not provide sufficient evidence about the OPP pathway supplies a larger portion of NADPH for biosynthesis, as light intensities in the experiments were not low enough.

RESPONSE: Llight intensity in the E. huxleyi, I. galbana and R. lamellosa experiments was low at 60 umol photons m-2 s-1, which we think is low enough to provide information about OPP supplying more NADPH for biosynthesis. Previous experiments and field studies have shown larger variation in fractionation at light intensities below 200
(van der Meer et al., 2015; Wolhowe et al., 2015), which might be caused by larger input of OPP derived NADPH. We will make this section more concise in a revised version.

There are quite a few different families (species) of halophytes. Just wonder if alkenones might be biosynthesized in different organelle among different species. The authors cited (Rontani et al., 2006) to suggest that alkenones are synthesized from these shorter chain fatty acids by elongation and subsequent decarboxylation in the chloroplast (Page 10, Lines 10-12), but then claimed "Chain elongation leading to longchain alkenones does take place in the cytosol" (Page 14, Line 4). Previous studies did show different D/H fractionation in biomarkers biosynthesized among different organelle. Would it be possible for difference in alkenone D/H fraction among different families due to different organelle for synthesis of alkenones?

RESPONSE: Fatty acids are formed and initially elongated in the plastid. Subsequent elongation occurs in the endoplasmic reticulum and utilizes a cytosolic pool of acyl-CoA (Huerlimann and Heimann, 2013). We will rephrase these two statements to emphasize that elongation takes place in the E.R., not the chloroplast, and relies on cytosolic intermediates. We agree strongly that different organelles impart a different biosynthetic signature / effect of fractionation and that is one of the main arguments of this manuscript (see page 11 lines 15-16, final line of page 11 to lines 1-2 of page 12, page 14, lines 7 - 12). It could be that different species, or species from the different Groups II and III, use different organelles for synthesis, but instead, a more plausible explanation for the species offset might be due to their osmotic regulation capacities.

Technical corretions 1. Replace "metabolic" with "biosynthetic" as the paper only address about the biosynthesis of biomarkers.

RESPONSE: We will fix this by changing the title of our manuscript to, "Impact of lipid biosynthesis pathways and growth parameters on hydrogen isotope ratios of hapto-phyte lipids".
2. Page 5, lines 8 and 12: "n-alkanes", here "n" should be italic RESPONSE:We will fix this.

3. Page 6, Line 22—-There was no Fig. 2c

RESPONSE: We will fix the figure notations for this section. Figure 2 shows fractionation plotted against nutrient concentration (a) and growth rate (b).

**BGD**
Fig. 1. Growth curves for the Temperature experiment based on chlorophyll fluorescence.

**BGD**